

# Interactive Impacts of Fire and Vegetation Dynamics on Global
# Carbon and Water Budgets using Community Land Model
# version 4.5
Hocheol Seo[1], and Yeonjoo Kim[1]
[1]Department of Civil and Environmental Engineering, Yonsei University, Seoul 03722, Korea.
*Correspondence to*: Yeonjoo Kim (yeonjoo.kim@yonsei.ac.kr)
**Abstract**
Fire plays an important role in terrestrial ecosystems. The burning of biomass affects carbon and water fluxes and the
distribution of vegetation. To understand the effect of the interactive processes of fire and ecological succession on
land surface carbon and water fluxes, this study utilized the Community Land Model version 4.5 to conduct a series
of experiments that included and excluded fire and dynamic vegetation processes. Results of the experiments that
excluded dynamic vegetation showed a global increase in net ecosystem production (NEP) in post-fire regions, which
has been shown in previous studies with the similar modeling practices. However, inclusion of dynamic vegetation
revealed a fire-induced decrease in NEP in some regions. Additionally, the carbon sink in post-fire regions reduced
when the dominant vegetation type was changed from trees to grasses. This study shows that inclusion of dynamic
vegetation enhances carbon emissions from fire by reducing terrestrial carbon sinks; however, this effect is somewhat
mitigated by the increase in terrestrial carbon sinks when dynamic vegetation is not used. Results also show that fire-
induced changes in vegetation modify the soil moisture profile because grasslands are more dominant in post-fire
regions; this results in less moisture within top soil layers compared to non-burned regions, even though transpiration
is reduced overall. These findings are different from those of previous fire model evaluations, that ignore vegetation
dynamics, and thus highlight the importance of interactive processes between fire and vegetation dynamics,
particularly when evaluating recent model developments with respect to fire and vegetation dynamics.
**Keywords**
Fire model, Dynamic vegetation model, Terrestrial carbon balance, Community Land Model, Terrestrial water balance



## 1 Introduction

Wild fire is a natural process that influences ecosystems and carbon and water cycles worldwide (Gorham, 1991; Bowman et al., 2009; Harrison et al., 2010). Climate and vegetation control both the occurrence of fire and its spread, which in turn affects climate and vegetation (Vilà et al., 2001; Balch et al., 2008), and when fire destroys forests and grasslands the distribution of vegetation is affected (Clement & Touffet, 1990; Rull, 1999). Fire causes the formation of trace gases and aerosols, which are important elements in the radiative balance of the atmosphere (Scholes et al., 1996; Fiebig et al., 2003); aerosols can affect surface air temperature, precipitation, and circulation (Tarasova et al., 1999; Lau & Kim, 2006; Andreae & Rosenfeld, 2008).

Changes in soil properties occur in regions affected by fire, and leaves and roots can be annihilated in such regions (Noble et al., 1980; Swezy & Agee, 1991). Each year, fire transports approximately 2.1 Pg of carbon from soil and vegetation into the atmosphere in the form of carbon dioxide and other carbon compounds (van der Werf et al., 2010). Harden et al. (2000) postulated that approximately 10–30% of annual net primary productivity (NPP) disappeared through fires in upland forests. In addition, transpiration and canopy evaporation decreases due to the reduction in leaf numbers (Clinton et al., 2011; Beringer et al., 2015). Soil has also been found to develop a water repellent layer during fire, which is attributed to intense heating (DeBano, 1991), and the ash produced by biomass combustion can impact the quality of runoff (Townsend & Douglas, 2000).

In post-fire regions, plant distribution gradually changes over time from bare ground to grassland, shrubland, and finally to forest during the process of ecological succession (Prach & Pyšek, 2001). In this respect, the structure and distribution of vegetation can be altered by fire in post-fire regions (Wardle et al., 1997); for example, several studies have suggested that the existence of grass and trees in the savanna can be attributed to fire (Hochberg et al., 1994; Sankaran et al., 2004; Baudena et al., 2010). However, fire can also wipe out succession.

Fire affects many different aspects of the Earth system. Therefore, a certain degree of process-based representation of fire is included in Earth system models, such as within dynamic global vegetation models (DGVMs), land surface models (LSMs), and Earth system models (ESMs; Rabin et al., 2017). Studies have applied fire models to global climate models to investigate the occurrence and spread of fire and how it impacts climate and vegetation (e.g., Pechony & Shindell, 2010; Li et al., 2012; 2013). For example, Bond et al. (2005) used the Sheffield DGVM (SDGVM) and performed the first global study on the extent to which fire determines global vegetation patterns by preventing ecosystems from achieving potential height, biomass, and dominant functional types expected under an ambient climate (i.e., potential vegetation).

In recent years, global fire models have grown in complexity (Hantson et al., 2016), and different fire models parameterize different impact factors such as fuel moisture, fuel size, the probability of lightning, and human effects. In this respect, the Fire Model Intercomparison Project (FireMIP) evaluated the strength and weakness of each fire model by comparing the performance of different fire models and suggesting improvements for individual models (Rabin et al., 2017).

A process-based fire parameterization of intermediate complexity known as the Community Earth System Model (CESM) has been developed and assessed within the framework of the National Center for Atmospheric Research (NCAR) (Li et al., 2012; 2013; 2014), and the latest satellite-based Global Fire Emission Database version



3 (GFED3), which is derived from moderate resolution imaging spectroradiometer (MODIS) fire count products, has
been used to improve fire parameterizations. Furthermore, the impact of fire on carbon, water, and energy balances
has also been investigated within the CESM framework (Li et al., 2014; Li & Lawrence, 2017). However, although
these studies have considered land-atmosphere interactions with CLM coupled to the atmospheric model, they have
ignored changes in global vegetation patterns due to fire processes, even though the initial model developed by Li et
al., (2012) was designed to consider vegetation dynamics (i.e., changes in vegetation distribution) within the
Community Land Model (CLM)-DGVM.

It is important to understand the possible influences of fire processes on water and carbon exchanges and

vegetation distribution, and their combined effects; however, few studies to date have assessed this complicated global
process. Therefore, in this study, we aim to understand the interactive effects of fire and ecological succession on
carbon and water fluxes at the land's surface. Specifically, we conduct a series of numerical experiments using the
NCAR CLM that variously include and exclude fire and dynamic vegetation processes. Our results show that the
impact of fire on carbon and water balances (especially in net ecosystem production (NEP) and soil moisture) on
ecological succession is different from that on static vegetation.

## 79    2 Model and Experimental Design

### 80    2.1 Model description

This study used CLM version 4.5, which is the land model of NCAR CESM version 1.2 The CESM is maintained by
NCAR's Climate Global Dynamics Laboratory (CGD), and it comprises different components such as land,
atmosphere, ocean, land ice, and ocean ice (Worley at el., 2011; Kay et al., 2012). Each component utilizes various
formulae to represent the complex interplay of physical, chemical, and biological processes, and each can be used
either independently or coupled (Smith et al., 2010; Neale et al., 2012; Bonan et al., 2013). Land surface in the CLM
is represented by sub-grid land cover (glacier, lake, wetland, urban, or vegetated), and vegetation coverage is
represented by 17 plant functional types (PFTs) comprising 11 tree PFTs, two crop PFTs, three grass PFTs, and bare
ground. For a detailed description of the model, please refer to Lawrence et al. (2011).

In this study, CLM 4.5 was extended using the biogeochemistry (BGC) model option; this configuration

simulates the carbon and nitrogen cycles in addition to biophysics and hydrology (Paudel et al., 2016). In CLM with
BGC, the spatial distribution of PFTs is set using monthly climatological satellite data (Lawrence & Chase, 2007),
which differs between months but not between years. Climatological PFT data are conserved based on MODIS and
Advanced Very High-Resolution Radiometer data. Land fractions are divided into bare ground, grass, shrub, and
evergreen/deciduous tree types. In addition, grass, shrub, and tree PFTs are classified into tropical, temperate, and
boreal types, based on the physiology and climate rules of Nemani et al. (1996). Vegetation is further divided into C3
or C4 plants based on MODIS derived leaf area index (LAI) values and the mapping methods of Still et al. (2003).

Certain BGC simulations were run using dynamic vegetation (DV; BGC-DV), which can simulate

biogeographical changes in the natural vegetation distribution and mortality processes (Castillo et al., 2012; 2013).

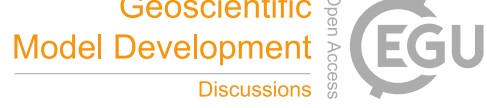

However, other BGC simulations did not employ the DV option (hereafter, BGConly). In BGC-DV simulations, a
PFT can occupy a region or degenerate by competing with other PFTs, or they can coexist under various environmental
factors, such as light, soil moisture, temperature, and fire (Zeng, 2010; Song & Zeng, 2013). In BGConly, whole-plant
mortality is parameterized by assuming an annual mortality rate of 2%. Conversely, plant mortality in BGC-DV is
determined by heat stress, fire, and growth efficiency (Rauscher et al., 2015).
In the fire model (Li et al., 2012, 2013; Bonan et al., 2013), fire is divided into four sections: non-peat fires
outside cropland and tropical closed forests, agricultural fires, deforestation fires in tropical closed forests, and peat
fires. Fire counts are determined based on natural and artificial ignitions, fuel availability, fuel combustibility, and
anthropogenic and unsuppressed natural fires related to socioeconomic conditions, and the burned area is calculated
by multiplying the fire count with the average fire spread, which is considered to be driven by wind speed, PFT, fuel
wetness, and socioeconomic influences. In other words, the burning and spread of fire are related to the CLM input
parameters of climate and weather conditions, vegetation conditions, socioeconomic conditions, and population
density.
Once the burned area is identified, fire impacts, including vegetation mortality, peat burning, and the carbon
cycle, can be addressed. For example, the amount of carbon emitted from the fire ($E$) is calculated as follows,
$$E = A \cdot C \cdot CC, \tag{1}$$
where $A$ is the burned area; $C$ is a vector with elements including the carbon density of the leaf stem and root, and
transfer and storage of carbon; and $CC$ is the corresponding combustion completeness factor vector. When the DV
option is included in the model, individual plants are killed by fire. The number of PFT individuals killed by fire
($P_{distrub}$) is calculated by,
$$P_{distrub} = \frac{A_b}{f A_g} P \, \xi, \tag{2}$$
where $P$ is the population density for each PFT, $\xi$ is the whole-plant mortality factor for each PFT, $A_g$ is the grid cell
area, $A_b$ is the burned area of each PFT, and $f$ is the fraction of coverage of each PFT.
The terrestrial carbon balance is affected when biomass is burned. In this respect, the net ecosystem exchange
(NEE) can be estimated using NEP and carbon loss due to biomass burning ($C_{fe}$) as follows,
$$NEE = -NEP + C_{fe}, \tag{3}$$
**2.2 Experimental design**
A series of global numerical experiments were conducted for this study using a spatial resolution of 1.9° longitude ×
2.5° latitude. Global climate data from the Climate Research Unit (CRU)-National Centers for Environmental
Prediction (NCEP) reanalysis were used for atmospheric driving forcing of CLM. Data from 1901 to 2000 included
6-h precipitation, air temperature, wind speed, specific humidity, longwave radiation, and shortwave radiation ranging.
Figure 1 summarizes the experimental process used in this study. Initial conditions for the 1850 equilibrium state were
provided by NCAR and used to simulate the 20th century transient run. The amount of atmospheric carbon dioxide
has increased since the onset of the Industrial Revolution in 1850, and the composition of land cover and vegetation



have changed (Vitousek et al., 1997; Pitman et al., 2004). Therefore, these changes need to be reflected when running
a 20th century transient simulation, and the final surface conditions should represent those of the year 2000 after
running the transient simulation using the CLM-BGC model.
Using the simulated surface conditions for 2000, four different 200-y equilibrium CLM simulations
(BGConly and BGC-DV simulations with and without the fire model) were conducted, as shown in Table 1. For
BGConly runs, a restart file from the transient run was used with and without the fire model (hereafter, BGConly-F
and BGConly-NF, respectively). Similarly, the BGC-DV runs were performed using the same restart file to simulate
the potential vegetation in 200-y offline BGC-DV runs both with and without the fire model (hereafter, BGC-DV-F
and BGC-DV-NF, respectively; Erfanian et al., 2016). In these simulations, the initial global land state was bare
ground (there were no plants) and soil conditions, such as soil moisture and temperature, were adjusted to those of the
year 2000 (Qiu & Liu, 2016; Wang et al., 2016). While the fire model is optional when using CLM with the BGC, it
is always run when using CLM with BGC-DV. Hence, the model was modified when conducting the BGC-DV-NF
run, and the burned area was set to equal zero to ignore any incidences of fire.
A comparison between the BGConly-F and BGConly-NF runs enables isolation of the impact of fire on the
land's surface, regardless of DV. In addition, the impact of fire and the interactive impact of fire and the distribution
of vegetation on the Earth's system can be identified by comparing BGC-DV-F and BGC-DV-NF runs. It is important
to remember here that this study focuses on the impact of fire and vegetation dynamics on land carbon and water
fluxes by forcing the CLM with the CRU-NCEP climate data (1991–2000) without consideration of land-atmosphere
feedbacks. Simulations were run for 200-y from the initial surface conditions of 2000 to derive potential land surface
conditions. In addition, the average surface conditions of the last 30-y were compared with the results of simulations,
and an analysis of this is presented in the following section.

## 3 Results and Discussions

### 3.1 Burned area

The simulated burned area from the BGC-DV-F and BGConly-F runs were compared to the GFED3 dataset, which is
a reanalyzed dataset derived from MODIS (Giglio et al., 2013). While GFED3 suggests a global burned area estimate
of 380 Mha/year for 1999 to 2011, the BGC-DV-F and BGConly-F runs show burned areas of 320 and 487 Mha/year,
respectively. Although these are comparable with GFED3, they are not identical because the model estimate run using
CRU-NCEP forcing is a 30-y average of the potential vegetation. In addition, agricultural fires are excluded in BGC-
DV-F, as it only simulates natural vegetation (Castillo et al., 2012).
It is apparent that BGC-DV-F estimates the GFED3 burned area more accurately than BGConly-F, and this
could be attributed to the fact that the fire model of Li et al. (2012) was originally developed using a comparison of
BGC-DV-F CLM simulations with GFED3. In a previous study using a BGC-F type simulation coupled to CAM (Li
& Lawrence, 2017), the annual burned area was found to be 489 Mha, which is similar to that of BGConly-F (487
Mha).



Spatial distributions of burned areas are compared in Figure 2. Compared to GFED3, both BGConly-F and

BGC-DV-F runs overestimate the burned area in the Americas and in Asia, while BGC-DV-F also underestimates the

burned area in Africa and Oceania. These results can be attributed to the differences between BGC-DV-F and

BGConly-F vegetation distributions, as shown in Figure 3 (where PFTs, excluding two crop PFTs, are simplified into

six vegetation groups; broadleaf evergreen trees, needleleaf evergreen trees, deciduous trees, shrubs, grasses, and bare

ground) (Rauscher et al 2015). In BGC-DV-F (Figure 3a), evergreen and deciduous trees show limited growth whereas

grasses and bare ground predominate in regions such as southern Africa. Overall, BGC-DV-F simulates trees on 37.5%

of the global land area; however, observations (Figure 3b) indicate that trees cover 41.46% (Table 2). However, more

trees provide increased fuel for the occurrence and spread of fire in BGC-DV-F compared to BGConly-F

**3.2 Interactions between vegetation and fire processes**
In this section, we assess the impact of fire processes on the distribution of vegetation by comparing BGC-DV-F and
BGC-DV-NF simulations (Table 2 and Figures 3 and 4). Figure 4 shows the vegetation distribution of BGC-DV-NF
and BGC-DV-F minus BGC-DV-NF;  difference plots clearly indicate large differences in vegetation cover in areas
of high fire frequency (i.e., South Africa, South America, western North America, India, and a portion of China), as
shown in Table 2, whereas areas with relatively low fire occurrence (i.e., the Arctic and desert regions) show small
differences.

The relationship between vegetation distribution and fire occurrence is investigated by estimating the fraction

of burned areas (Figure 5), where fractions are grouped into four categories (>10%, 10%~1%, 1%~0.1% and, <0.1%)
for each vegetation type, and they illustrate a nonlinear change in vegetation distribution in response to post-fire area.
Changes in the vegetation distribution are small in areas with minimal fire occurrence or where the burned area fraction
is small (0.1~1%). However, relatively large changes in vegetation distribution are apparent when the burned area
fraction exceeds 1%. Furthermore, there are large changes in the vegetation distribution in areas with burned area
fractions above 10%, including increases in bare ground, grass, and shrubs (31.19, 52.28, and 7.91%, respectively)
but decreases in deciduous, needleleaf evergreen, and broadleaf evergreen trees (8.85, 79.22, and 91.17%,
respectively).

Areas that experience a higher frequency of fire occurrence have larger vegetation distribution differences,

which suggests that fire has an influence on vegetation mortality. In ecological processes, plants die in regions where
fire occurs; grasses with rapid growth rates then occupy regions after fire. Therefore, fire increases the ratios of bare
ground and grassland but reduces the percentage number of trees. However, there are no marked changes in the
fractions of shrubs and deciduous trees in the middle of the ecological succession process with respect to the presence
or absence of fire (Table 2). When fire occurs in a region where shrubs grow, the ratio of shrubland is diminished, but
fire increases the ratio of shrubland in regions where trees may evolve from shrubs. In the same way as shrubs, the
deciduous trees are increased or decreased due to fire. Thus, it is apparent that the role of fire in areas of shrubland
and deciduous trees differs according to the region, and the actual vegetation distribution is a result of complicated
factors that include fire, climate, topography, and soil conditions (He et al., 2007; Cimalová & Lososová, 2009).



### 3.3 Fire impact on carbon balances

The direct and indirect impacts of fire on carbon balances were investigated by exploring the difference between fire impact when using state and dynamic vegetation (Figure 6 and Table 3). The impact of fire in two cases (BGConly-F minus BGConly-NF (BGCOnly) and BGC-DV-F minus BGC-DV-NF (BGC-CV) were estimated by averaging the final 30-y of each 200-y simulation.

Carbon emissions due to fire (direct impacts) are shown in Figure 6. The spatial distributions of the BGConly and BGC-DV runs are similar, but average annual emissions are higher in BGConly (3.4 Pg) compared to BGC-DV (3.0 Pg). This result could be attributed to trees being less dominant in BGC-DV compared to BGConly, which thus causes a reduced fuel load.

We note that estimates of carbon emissions from BGConly and BGC-DV are relatively high; however, they do fall within the range of previous findings. For example, 1999–2011 GFED3 data estimated annual direct carbon emissions as being approximately 2.0 Pg. Furthermore, Mouillot et al. (2006) estimated annual carbon emissions as being approximately 3.0 Pg for the end of 20th century and approximately 2.5 Pg for the 20th century average; and Li et al. (2014) and Yue et al. (2015) both estimated 20th century emissions as being 1.9 Pg C yr$^{-1}$ using the CLM4.5 and ORCHIDE land surface models, respectively.

In addition to direct carbon emissions, fire influences terrestrial carbon sinks by impacting ecosystem processes, as shown in Figure 6. Fire increases the NEP in post-fire regions in BGConly simulations, which is consistent with the findings of previous studies (Li et al., 2014). However, the overall NEP decrease is 2.5 Pg C y$^{-1}$ in this study, which is greater than the value of 1.9 Pg C yr$^{-1}$ determined by Li et al. (2014). However, Li et al. (2014) performed a transient simulation from 1850 to 2004, whereas the BGConly runs in our study were conducted following an equilibrium simulation using 2000 as the reference year, which thus meant that no fire exchanges were due to land cover changes.

Simulations that ignore vegetation dynamics (i.e., the BGConly runs in this study; Li et al., 2014; Yue et al., 2015) show a global fire-induced NEP increase when comparing fire-on and fire-off runs. However, a decrease in fire-induced NEP is apparent in some regions when using BGC-DV (Figure 6). This carbon sink reduction occurs in regions where dominant PFTs changed from broadleaf and needleleaf evergreen trees to grasses (as shown in Table 3 and Figure 6). Table 4 shows the correlation coefficients between percentage changes in vegetation types and changes in carbon fluxes (NEP, NPP, and heterotrophic respiration (Rh)) for six different PFTs in each grid cell), and Figure 7 shows the broadleaf evergreen tree, needleleaf evergreen tree, and grass PFTs. It is apparent that NEP changes are strongly linked to changes in the dominant PFTs; for example, decreases in broadleaf evergreen and needleleaf evergreen trees, and increases in grasses. Furthermore, associations between changes in NEP and PFTs are related to changes in both NPP and Rh to some extent. Our results differ from those of previous studies that did not consider vegetation dynamics (e.g., Amiro et al., 2010), because the inclusion of vegetation dynamics enables the model to capture NEP decreases in post-fire regions at the beginning of post fire-succession.

As land use change is not considered in this study, the overall impact of fire was estimated by the sum of carbon emissions and terrestrial carbon sinks (Eq. 3). Both simulations resulted in carbon sources in the post-fire regions, even though different processes were involved. Although carbon emissions due to fire were partly negated





by the increased terrestrial carbon sinks in the BGConly runs, they were enhanced by the reduction in terrestrial carbon
sinks in the BGC-DV runs.

### 3.4 Fire impact on water balances

The impact of fire on the water balance was examined by estimating changes in runoff, evapotranspiration, and soil
moisture, and by making a comparison between BGConly and BGC-DV (Table 5 and Figure 8). Increases in runoff
and decreases in evapotranspiration (ET) were found in post-fire regions to different degrees, which is consistent with
the results of previous studies (Neary et al., 2005; Li & Lawrence, 2017). Our study used CLM as a standalone model
without coupling to the atmospheric or ice models, whereas Li and Lawrence (2017) examined the impact of fire on
the global water budget using CLM-BGC coupled with the CAM and CICE models and found that the impact of fire
on global annual precipitation was limited.
Li and Lawrence (2017) pointed out that a reduction in the vegetation canopy (LAI; Table 6) is a critical
pathway for fire impacting on ET and leading to its decrease. Fire events lower the leaf area, which decreases
vegetation transpiration and canopy evaporation; however, they also expose more of the soil to the air and sunlight,
which increases soil evaporation. Post-fire decreases in vegetation height (Table 6) can both increase and decrease ET,
as the resulting decrease in land surface roughness potentially reduces water and energy exchanges and leads to higher
leaf temperatures and wind speeds. In this study, both BGConly and BGC-DV runs show the vegetation canopy is the
main pathway leading to ET decrease, which is similar to the findings of Li and Lawrence (2017). In addition, an
examination of how changes in the vegetation composition within post-fire regions influences the above mechanisms
shows that overall impacts of ET and runoff do not differ greatly when dynamic vegetation is employed by the model.
However, results show that fire-induced vegetation changes (from trees to grass or bare ground) in BGC-DV lead to
a marked decrease in canopy transpiration and increased soil evaporation relative to BGConly. Fire destroys plant
roots and leaves, and changes in the dominant vegetation types in BGC-DV lead to changes in the soil moisture profile
through reduced transpiration (Figure 9 and Table 7). Consequently, there is less water stress in each soil layer within
burned areas than in non-burned areas. Grasslands dominate in post-fire regions when using BGC-DV, and they absorb
and transpire more water from the top soil layer compared to trees (Mazzacavallo & Kulmatiski, 2015). There is thus
less moisture in the top soil layers in fire affected regions than in non-burned regions, despite the fact that the overall
transpiration is diminished. Put simply, fire has an impact on the vegetation distribution, which in turn impacts the
soil water profile.
Changes in ET and runoff do not differ markedly between BGConly and BGC-DV, despite differences in the
vegetation canopy and height, and soil moisture. This result could be attributed to the fact that an offline CLM was
used, which does not allow for land-atmosphere interactions. We therefore expect that the impact of fire on
precipitation would be more significant in BGC-DV than in BGConly because fire directly influences land cover
characteristics.



### 4. Conclusions


To understand the interplay of vegetation dynamics and fire impacts, we conducted a series of numerical experiments
using CLM both with and without fire and dynamic vegetation processes enabled. In particular, we investigated fire
influences on vegetation distribution, and how such changes influence terrestrial carbon and water fluxes.
As expected, results showed that fire interrupts the process of ecological succession, which thus impacts the
global vegetation distribution. Fire transforms some regions into bare ground, and grasses quickly dominate as they
grow faster than trees. For shrubs and deciduous trees in the mid-stages of ecological succession, we found no large
differences in the overall coverage ratios between simulations including vegetation dynamics and those that did not.
Simulations that did not consider vegetation dynamics showed a fire-induced global increase in NEP; however, a fire-
induced decrease in NEP was found in some regions in BGC-DV. We also found a carbon sink reduction in regions
where the dominant PFT changed from broadleaf and needleleaf evergreen trees to grasses. While carbon emissions
due to fire were partly negated by increased terrestrial carbon sinks (NEP) in BGConly runs, they were enhanced by
the reduction in terrestrial carbon sinks in BGC-DV runs when dynamic vegetation was considered.
Fire-induced changes in vegetation from trees to grass or bare ground resulted in a decrease in canopy
transpiration and increased soil evaporation in post-fire regions for BGC-DV; however, there were no marked
differences in the overall impacts on ET and runoff between simulations that used dynamic vegetation and those that
did not. Interestingly, however, changes in dominant vegetation types in BGC-DV led to changes in the soil moisture
profile. Furthermore, the increased distribution of grassland cover was more dominant in post-fire regions, which then
resulted in less moisture in the top soil layers compared to non-burned areas, despite that fact that transpiration
diminished overall.
Enabling the vegetation dynamics module in the CLM assists in gaining an understanding of the interactive
impacts of fire and vegetation dynamics. However, uncertainty still exists because of the limited simulated potential
vegetation distribution using CLM with BGC-DV-F. Furthermore, the final potential vegetation state of the BGC-DV
model did not always correspond to the observed distribution (Figure 3). For example, shrubs in the tundra were found
to be rare in both BGC-DV-F and BGC-DV-NF. Furthermore, crops, needleleaf evergreen boreal, and shrub boreal
cannot be simulated by the DV module, as noted in previous studies (Zeng et al., 2008).
The fire module in CLM is parameterized to estimate fire occurrence, fire spread, and fire impact. Thresholds
used to estimate fuel combustibility are dependent on relative humidity and surface air temperatures; however, these
values may not be suitable for all regions (Zhang et al., 2016). In addition, the economic impact of fire occurrence and
the socioeconomic impact of fire spread are estimated using the input datasets of population density (person/km$^2$) and
GDP (US$/capita), respectively (Li et al., 2013). Uncertainty due to socioeconomic factors should be noted for both
historical and future simulations, because changes in these factors may vary by country (Steelman & Burke, 2006). It
is evident that our understanding of fire needs to improve, because fire plays an important role in the distribution of
vegetation and in carbon, water, and energy cycling. Furthermore, this study shows that fire models are strongly
impacted by vegetation distribution; therefore, fire simulations would improve if improvements were made to the
dynamic vegetation model.

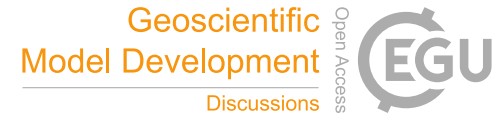

**Code and Data Availability**

The code of and input datasets for CLM are downloaded from the NCAR CLM website (refer to cesm.ucar.edu).

**Author Contribution**

YK and HS designed the study and HS performed the model simulations with processing the data and modifying the code. Both YK and HS analyzed the results and wrote the manuscript.

**Acknowledgements**

This study was supported by the Basic Science Research Program through the National Research Foundation of Korea, which was funded by the Ministry of Science, ICT & Future Planning (2018R1A1A3A04079419), and by the Korea Polar Research Institute (KOPRI, PN17900).

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





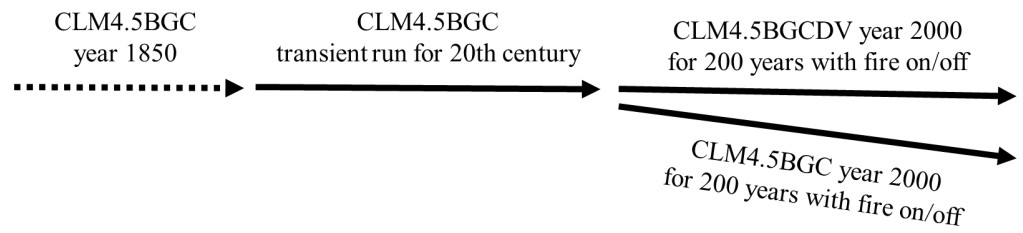


**Figure 1:** Flowchart showing model simulations conducted to investigate the interactive impact of fire and ecological succession on the Earth system using Community Land Model (CLM4.5) simulations extended with biogeochemistry (CLM4.5BGC) and BGC with dynamic vegetation (CLM4.5BGCDV).




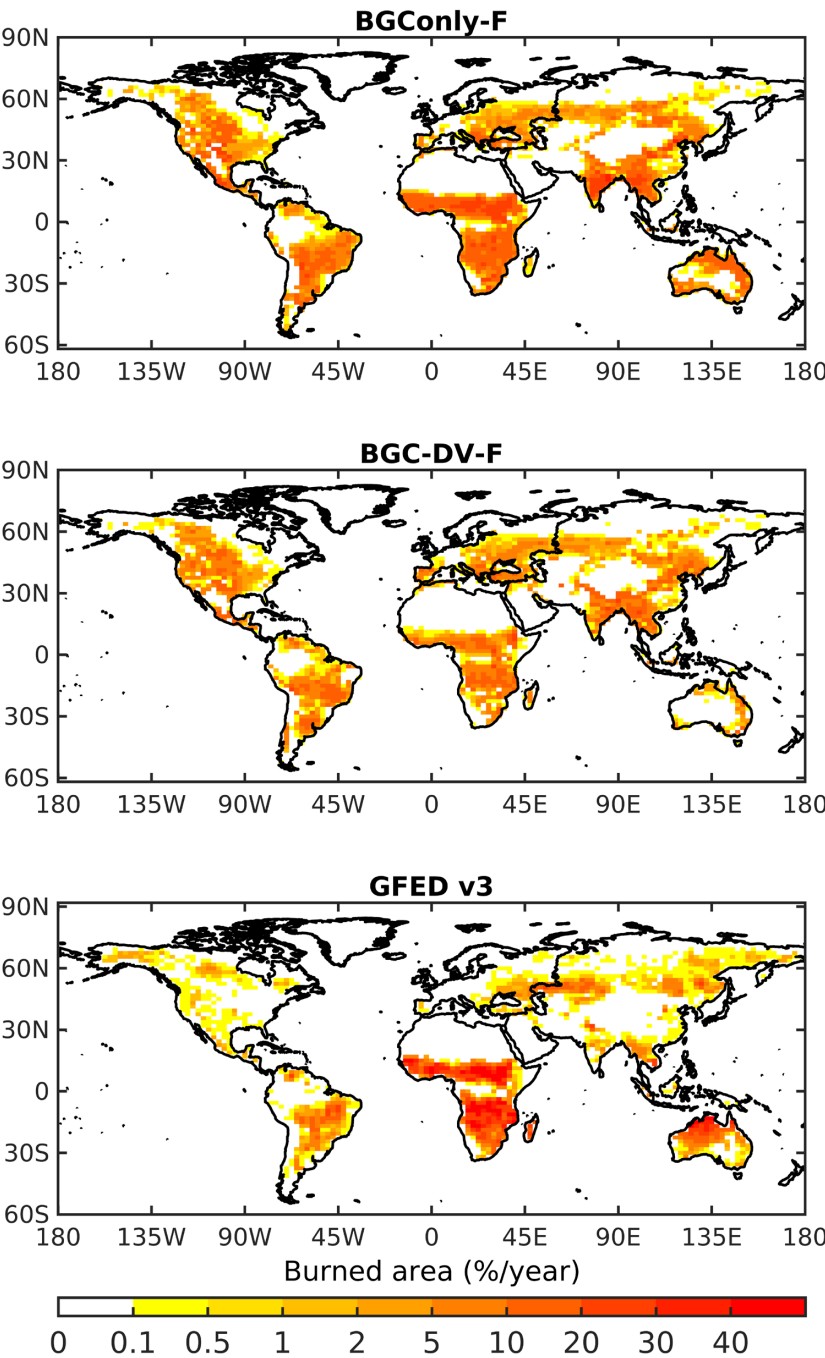

**Figure 2:** Annual burned area percentage by grid cell for CLM4.5BGC with fire (BGConly-F), CLM4.5BGCDV with fire (BGC-DV-F), and Global Fire Emission Database version 3 (GFED v3).

.



**Figure 3:** Percentages of land cover type (bare ground (BE), grass (GR), shrub (SH), deciduous (DE), needleleaf evergreen (NE), and broadleaf evergreen (BE)) in BGC-DV-F and BGConly.








**Figure 4:** Percentages of land cover type (bare ground (BE), grass (GR), shrub (SH), deciduous (DE), needleleaf evergreen (NE),

and broadleaf evergreen (BE)) in BGC-DV-NF and differences in plant cover between BGC-DV-F and BGC-DV-NF.

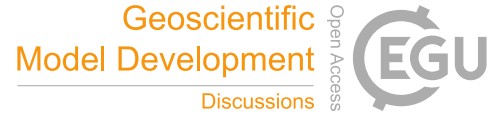



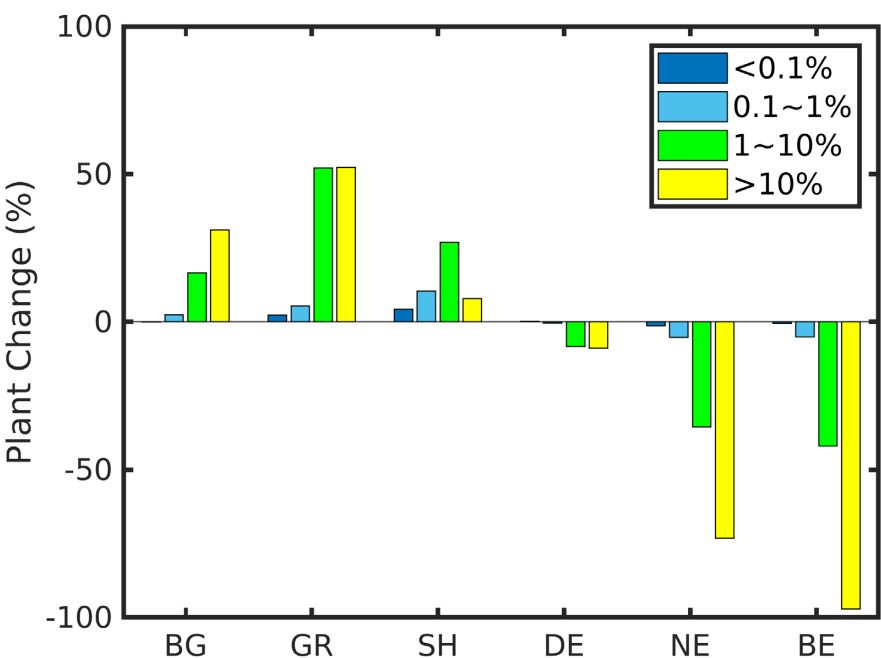


**Figure 5:** Changes in vegetation ratios for four burned area categories: under 0.1%, 0.1~1%, 1~10%, and greater than 10%.





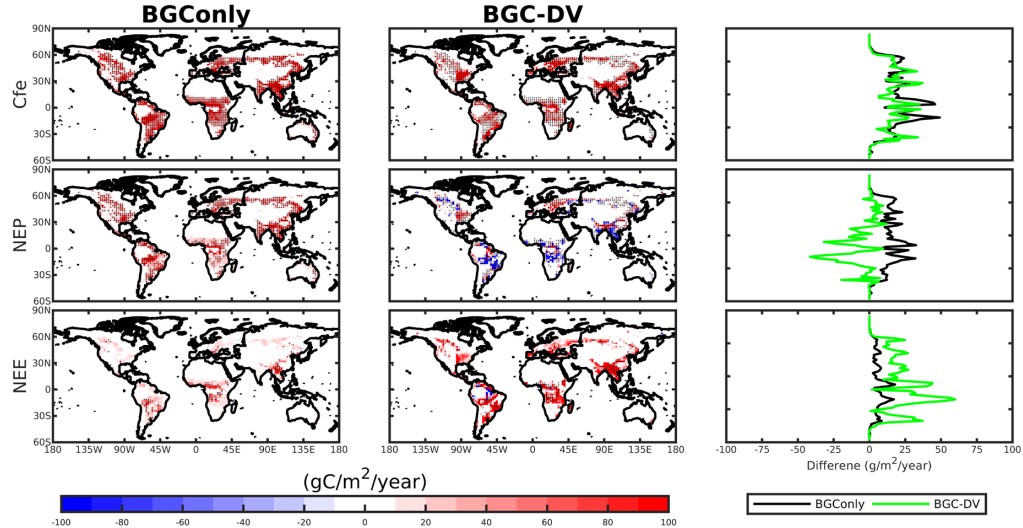

**Figure 6:** Differences in carbon emissions (Cfe), net ecological production (NEP), and net ecosystem exchange (NEE) due to fire
between BGC and BGC-DV. Hashed areas indicate that difference passed the Student's t-test at 0.05 significance level.
Latitudinal mean differences are plotted in far-right column.




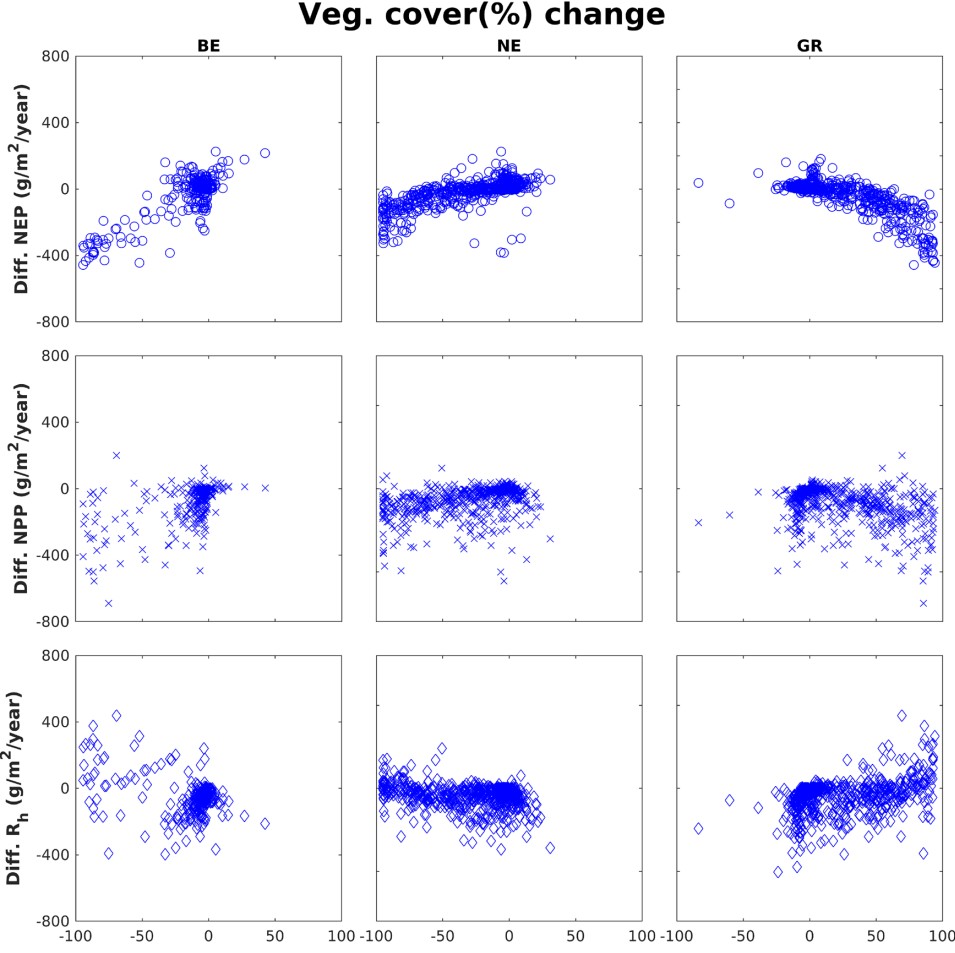

**Figure 7:** Response of differences (BGC-DV-F minus BGC-DV-NF) in NEP, NPP, and $R_h$ to percentage changes in bare ground (BE), needleleaf evergreen (NE), and grass (GR) vegetation types.





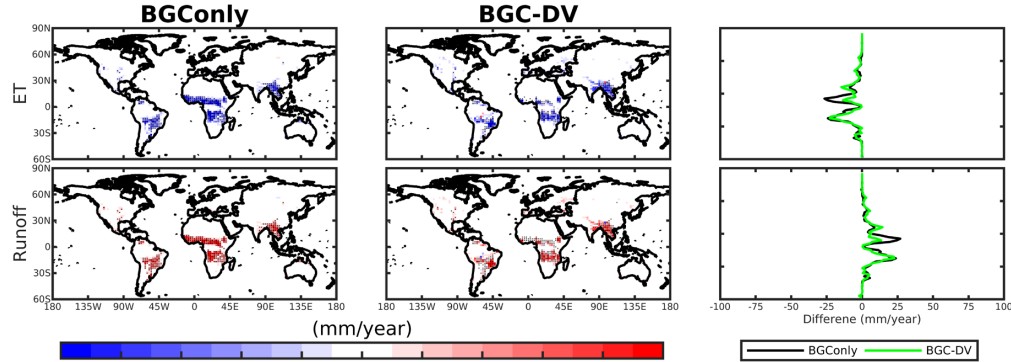

**Figure 8:** Same as Figure 6 but for water fluxes (evapotranspiration (ET) and runoff)



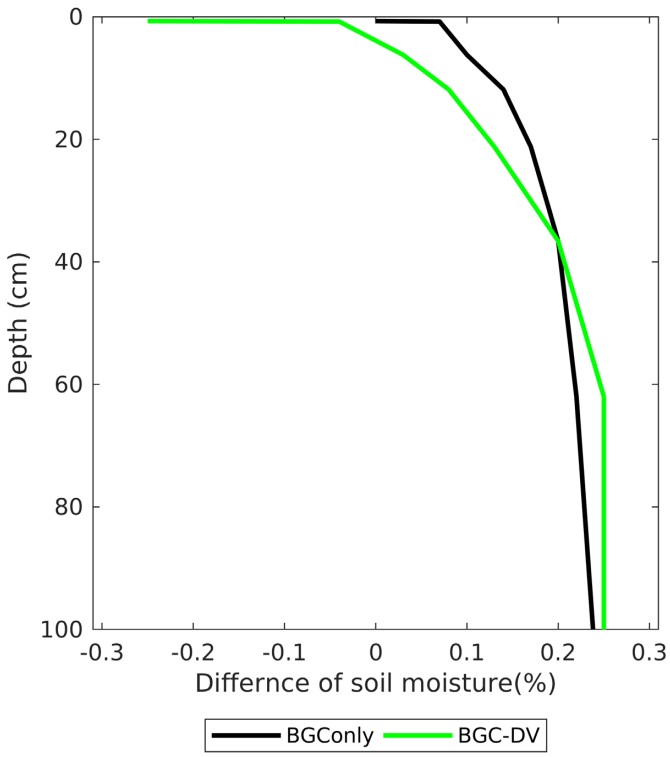


**Figure 9:** Difference in soil moisture (%) in BGConly and BGC-DV simulations.





**Table 1:** Configurations of four experiments used in study.

| Model option | | Dynamic vegetation model | |
|---|---|---|---|
| | | ON | OFF |
| Fire model | ON | BGC-DV-F | BGConly-F |
| | OFF | BGC-DV-NF | BGConly-NF |






**Table 2:** Percentage (%) land cover types (bare ground, grass, shrub, deciduous, needleleaf evergreen, and broadleaf evergreen)
in BGConly, BGC-DV-F, and BGC-DV-NF.

|  | BGConly | BGC-DV-F | BGC-DV-NF |
|---|---|---|---|
| Bare ground | 28.17 | 41.21 | 38.66 |
| Grass | 20.13 | 21.25 | 16.53 |
| Shrub | 8.41 | 4.75 | 4.24 |
| Deciduous | 12.78 | 12.29 | 12.67 |
| Needleleaf evergreen | 9.96 | 14.73 | 20.54 |
| Broadleaf evergreen | 10.31 | 5.73 | 7.33 |
| Crop | 10.25 | - | - |







**Table 3:** Annual means of carbon budget for GPP, NPP, Ra, Rh, NEP, NEE, and Cfe and differences (Fire on- Fire off) between
BGConly and BGC-DV simulations in Pg C yr-1. Asterisk (*) index indicates that difference in Fire on and Fire off simulations
passed the student's t test at $\alpha = 0.05$ significance level.

|  | BGConly | | | BGC-DV | | |
|---|---|---|---|---|---|---|
|  | Fire on | Fire off | Diff | Fire on | Fire off | Diff |
| $C_{fe}$ | 3.49 | 0.00 | 3.49* | 2.98 | 0 | 2.98* |
| GPP | 130.51 | 144.24 | -13.73* | 122.01 | 136.93 | -14.92* |
| NPP | 56.66 | 63.17 | -6.51* | 52.14 | 55.56 | -3.42* |
| $R_a$ | 73.85 | 81.08 | -7.23* | 69.87 | 81.37 | -11.50* |
| $R_h$ | 52.75 | 61.73 | -8.98* | 41.19 | 43.79 | -2.60* |
| NEP | 3.91 | 1.44 | 2.47* | 13.65 | 14.67 | -1.02* |
| NEE | -0.42 | -1.44 | 1.02* | -5.27 | -8.87 | 3.60* |






**Table 4:** Correlation coefficients between carbon fluxes (NEP, NPP, Rh) and percentage changes in vegetation cover for
broadleaf evergreen (BE), needleleaf evergreen (NE), deciduous (DE), shrub (SH), grass (GR), and bare ground (BG).

|          | BE    | NE    | DE    | SH    | GR    | BG    |
|----------|-------|-------|-------|-------|-------|-------|
| NEP      | 0.84  | 0.68  | 0.34  | -0.28 | -0.80 | -0.14 |
| NPP      | 0.56  | 0.44  | 0.34  | -0.30 | -0.47 | -0.35 |
| $R_h$    | -0.36 | -0.17 | -0.01 | -0.13 | 0.27  | -0.30 |






**Table 5:** Same as Table 3, but for the water budgets of ground evaporation (GE), canopy evaporation (CE), canopy transpiration
(CE), evapotranspiration (ET), and total runoff (RO) in 103 km$^3$ yr$^{-1}$.

|  | BGConly | | | BGC-DV | | |
|---|---|---|---|---|---|---|
|  | Fire on | Fire off | Diff | Fire on | Fire off | Diff |
| GE | 20.87 | 19.27 | 1.60* | 23.29 | 19.61 | 3.68* |
| CE | 15.71 | 16.39 | -0.68* | 15.62 | 16.88 | -1.26* |
| CT | 38.41 | 40.42 | -2.01* | 37.68 | 40.99 | -3.31* |
| ET | 74.99 | 76.08 | -1.09* | 76.59 | 77.48 | -0.89* |
| RO | 31.09 | 30.02 | 1.07* | 29.51 | 28.64 | 0.87* |







**Table 6:** Same as Table 3, but for in LAI ($m^2/m^2$) and vegetation height(m).

|  | BGConly | | | BGC-DV | | |
|---|---|---|---|---|---|---|
|  | Fire on | Fire off | Diff | Fire on | Fire off | Diff |
| LAI | 2.13 | 2.36 | -0.23* | 2.24 | 2.62 | -0.38* |
| Height | 7.05 | 7.45 | -0.4* | 6.03 | 7.76 | -1.73* |




**Table 7:** Same as Table 3, but for soil moisture in (%) at each soil depth.

| Depth | BGConly | | | BGC-DV | | |
|---|---|---|---|---|---|---|
| | Fire on | Fire off | Diff | Fire on | Fire off | Diff |
| 0.71 cm | 21.22 | 21.22 | 0.00* | 20.48 | 20.73 | -0.25* |
| 0.79 cm | 23.22 | 23.15 | 0.07* | 22.59 | 22.63 | -0.04* |
| 6.23 cm | 23.24 | 23.14 | 0.10* | 22.61 | 22.58 | 0.03* |
| 11.89 cm | 22.72 | 22.58 | 0.14* | 22.14 | 22.06 | 0.08* |
| 21.22 cm | 22.37 | 22.2 | 0.17* | 21.83 | 21.7 | 0.13* |
| 36.61 cm | 22.48 | 22.28 | 0.20* | 21.98 | 21.78 | 0.2* |
| 61.98 cm | 22.57 | 22.35 | 0.22* | 22.1 | 21.85 | 0.25* |
| 103.8 cm | 22.45 | 22.21 | 0.24* | 21.95 | 21.7 | 0.25* |
