# Peer review of "Interactive impacts of fire and vegetation dynamics on global"

_Geoscientific Model Development, 2018_

## Referee Comment (RC1) · Anonymous Referee #1 · 19 Oct 2018

[11pt]article [margin=1in]geometry

**General comments**

In this manuscript, Seo and Kim present the results of a study designed to assess the relative and interactive effects of simulating fire and dynamic vegetation on carbon and water cycling in the Community Land Model. One especially interesting finding is that fire seems to increase net ecosystem productivity, but only when dynamic vegetation is turned off. Many of the other results are not very novel, but are appropriate for

Geoscientific Model Development because they add evidence supporting existing findings, and could help to interpret future CLM experiments.

This work could be valuable for the large community of researchers using CLM, as well as for global vegetation and land system modelers in general. Unfortunately, certain experimental design choices, coupled with uncertain explanation of model run setups, render parts of this manuscript impossible to confidently evaluate. I thus recommend that this paper be **resubmitted with major revisions**.

**Specific comments**

The spinup and transient model runs need to be much better explained. Table 1 would have been a good place to clarify things, but as it is now that table does not really give any useful information. Here's the information I would like to see in a revised Table 1, along with the gaps left by the Methods text (and a read through of the Methods for Qiu and Liu 2016):

|  | CLM4.5BGC 1850 | CLM4.5BGC 20th cent. | BGConly | BGC-DV |
|---|---|---|---|---|
| Time | ??? | 1901–2000 | 200 years | 200 years |
| Climate forcings | ??? | 1901–2000 (CRU-NCEP) | Repeated 1901–2000 (CRU-NCEP) | Repeated 1901–2000 (CRU-NCEP) |
| $[CO_2]$ |  |  |  |  |
| Biogeog. shifts? | Yes? | Yes? | No (constant map) | Yes |
| Initial veg. | No (bare soil)? | Yes (as resulting from "CLM4.5BGC 1850")? | No (bare soil)? | No (bare soil)? |
| Initial soil | Uninitialized | As resulting from "CLM4.5BGC 1850" | As resulting from "CLM4.5BGC 20th cent." | As resulting from "CLM4.5BGC 20th cent." |
| Land use | ??? | ??? | At least crops? | No? |
| Fire | On? | On? | BGConly-F: On BGConly-NF: Off | BGC-DV-F: On BGC-DV-NF: Off |

- It is unclear exactly which runs were initialized with no vegetation (i.e., bare soil) because it is unclear what is being referred to by "In these simulations" on L 141. The idea that the BGConly and BGC-DV runs might be initialized with suddenly bare ground is concerning; this choice could have serious carbon cycling implications by itself. This should be justified, and well.

- If my interpretation is correct about the "Vegetation at beginning" row, how was the vegetation C present in 2000 removed for the start of the BGConly and BGC-DV runs? Was it removed from the land system entirely, or was it all killed and left to decompose?

- The use of climate forcings for 1991–2000 seems unwise. Generally, periods of at least 20–30 years are used in this sort of experiment, to better capture the full range of synoptic climate variability. It's especially egregious to use the 1990s specifically, because the 1998 ENSO event resulted in an extreme fire year.

- It is only explained near the end of the manuscript (LL 299–300) that crops are not simulated in the BGC-DV experiments. This, along with the "CR" panels in Fig. 3 (although CR is not defined anywhere), leads me to understand that crops are simulated in the BGConly experiments. But nowhere is there any information about (a) other land uses in those experiments, (b) land uses in the spinup and transient experiments, or (c) what is used instead of cropland in the BGC-DV experiments.

Unfortunately, the lack of clarity with regard to the model experiment setups makes confident appraisal of the rest of the manuscript impossible. I will attempt to assess what I can, couching my comments in the necessary uncertainty.

Section 3.1 (comparing simulated burned area with GFED3) is extremely problematic. Although I'm uncertain about the specifics of the experimental design, it seems clear that the runs are not intended as a way of reflecting reality but rather as an exploration of model mechanics. This is suggested by the use of equilibrium runs using 1991–2000 climate—a period in which the land system was certainly not in equilibrium, because of (among other factors) the continued recovery of forests in the northern hemisphere. Perhaps that's not an issue in these runs: It's possible that land use was turned off (there's no way to know, because it's not described in the Methods), but if that's the case, that's just another reason why a comparison of the model outputs to observations makes little sense. And even if one ignores all that, there's the problem that the simulations use 1991–2000 climate but the comparison is to burned area data from 1999–2011. The 1998 ENSO event resulted in an extreme fire year, which would be captured in the climate forcing (and ideally thus in the model

[Figure]

output) but not the observational data. The third paragraph of Sect. 3.3 (LL 213–218) is problematic for the same reasons.

Figure 3 (land cover comparisons between BGC-DV-F and BGConly) is confounded by the fact that crops were not simulated in the BGC-DV runs. What land cover is being simulated instead? Unless there is some kind of adjustment going on, the area that should be cropland is instead in some other land cover category in BGC-DV-F.

In Figure 5 and the discussion thereof (LL 184–192), it is unclear what is meant by "changes in the vegetation distribution." Does that refer to BGC-DV-F vs. BGConly-F, or BGC-DV-F vs. BGC-DV-NF? This makes it unclear how to interpret the results presented in the figure and text: Are we looking at an effect of including dynamic vegetation or of including fire?

The following, in Sect. 3.4, is incomplete: "Changes in ET and runoff do not differ markedly between BGConly and BGC-DV, despite differences in the vegetation canopy and height, and soil moisture. This result could be attributed to the fact that an offline CLM was used, which does not allow for land-atmosphere interactions." It actually also indicates that including dynamic vegetation doesn't make much difference for the physiological and physical processes of the land system governing evapotranspiration and runoff.

Other comments:

- The spinup and 20th century runs were performed with CLM4.5BGC, not CLM4.5BGC-DV. What input data were used for land use and vegetation?

- If changes were to be made to make Sect. 3.1 justifiable (see above), why would GFED3 be used instead of the more recent GFED4, or even better, GFED4s? This could change the interpretations in Sect. 3.1, for instance, since GFED4s

gives global burned area of 476 Mha/year—much closer to BGConly-F instead of BGC-DV-F.

- Tables 3 and 5: It is not clear what the t-tests are actually testing. Are they testing the difference of each experiment's mean difference from zero (i.e., the effects of including fire), or the difference between the two models' mean differences (i.e., the interactive effect of including dynamic vegetation and fire)?

- Throughout the paper, more effort should be made to distinguish the discussion of fire effects vs. dynamic vegetation effects.

**Technical corrections**

- L 58: Since the first FireMIP results paper has not been published, it would be more accurate to say "is evaluating" rather than "evaluated".

- L 64:

  – The most recent version of GFED is v4, not v3.
  – Because it's the name of a specific sensor rather than a general technology, Moderate Resolution Imaging Spectroradiometer should be capitalized.
  – In addition to MODIS fire counts, GFED also considers MODIS burned area.

- L 81: Period missing after "1.2".

- LL 301–302: "Thresholds used" should be "Thresholds are used".

- Fig. 3:

  – "Plant cover" should be simply "Coverage" or something similar, because bare ground by definition has no plants.

- – "BGConly" should be "BGConly-F".
  - – L 512: "bare ground (BE)" should be "bare ground (BG)". CR is not defined.
  - – L 513: "BGConly" should be "BGConly-F".
  - – L 515: "bare ground (BE)" should be "bare ground (BG)".
- Fig. 6: "Differene" should be "Difference".

---

## Referee Comment (RC3) · Anonymous Referee #2 · 9 Nov 2018

**General Comments**

This paper is evaluating the impact of incorporating fire and dynamic vegetation in the Community Land Model (CLM). The paper presents a clear description of the impact of including each of these processes by considering the change in burned area, vegetation cover, carbon balances (net ecological production and net ecosystem exchange) and water balances (evapotranspiration, runoff and soil moisture). The structure of the paper is logical and easy to follow.

The method employed for this study is a set of four experiments with and without dynamic vegetation and with and without fire, which seems a valid way of testing the

impact of each of these processes. However, while the paper starts well and includes a good introduction, the model description and experimental design needs more detail, and the analysis is ambiguous in several places. Specific comments on this are outlined below. The experiments here are not particularly novel, with fire and dynamic vegetation having been implemented in this model for several years as cited in the paper, but it is useful to have an evaluation of the impact of both processes as they are both important in land-surface and Earth System modelling. I recommend resubmission subject to addressing the following points.

Specific Comments

1) The BGConly model can be run with and without fire, and the results show that aspects of the vegetation (GPP, NPP, NEP etc, Table 3) are impacted when fire is included. But in the model description it says that the spatial distribution of PFTs is set using satellite data from MODIS and that a whole-plant mortality rate of 2% annually is assumed, rather than being determined by heat stress and fire etc. as it is in the dynamic vegetation model. So presumably the fire effects some aspects of the carbon cycle in the BGConly model, but not vegetation cover? I think this needs some clarification in the model description section – exactly what aspects are modified by including fire in the BGConly model, and what is not. It may be that this process is described in another paper, but it is necessary to understand this for the rest of this paper so it should be outlined here.

2) Related to the previous point, the BGConly-F results (Figure 3) are at one point referred to as 'observations' (line 175). This may be the case in terms of vegetation cover if this is prescribed and not altered by fire, but in the rest of the paper this is one of the experimental runs being evaluated, so it needs to be clearly stated that this exactly the same as the satellite data, both in BGConly-F and BGConly-NF, and it is therefore valid to treat this as observations. The labelling throughout the text is also quite confusing which doesn't help. For example the label 'BGConly' on Figure 3b doesn't specify if this is the fire on or off run, and 'BGConly' is first described as the

non-dynamic vegetation option (line 99) and then later as the impact of fire 'BGConly-F minus BGConly-NF' for figure 6 onwards (line 207).

3) The method for calculating NEP should be included, probably before equation (3) for NEE

4) The GFED3 dataset is used here for evaluating the burned area, but there are more up to date datasets now including small fires such as GFED4.1s. Is there a reason why GFED4.1s was not used here?

5) It seems strange that the original fire model was designed to consider vegetation dynamics (line 69) and the fire model is always run with BGC-DV (line 144), and that it simulates agricultural fire (line 105), but the DV model doesn't include crops (line 162). Presumably the agricultural fire function is only available in BGConly mode? This also needs explaining in the model description. And related, CR (I assume this is crop) is shown in figure 3 but not mentioned at all in the text, and is only available for BGConly so cannot be compared to BGC-DV – why include this if it isn't part of the analysis?

6) $\xi$ is the whole-plant mortality factor for each PFT (Line 120). What are the values of this factor and how is it determined?

7) Line 134 states that 'the final surface conditions should represent those of the year 2000 after running the transient simulation'. This is fine, but line 141 states 'In these simulations, the initial global land state was bare ground... and soil conditions... were adjusted to those of the year 2000'. Does this mean the initial land state was bare ground at the start of the transient simulation, not reset at 2000? I'm not sure why you would reset vegetation at 2000 after doing a transient run, so this is probably a wording issue, but then why adjust soil conditions to 2000? I would also have expected a spin-up at the beginning considering the initial state is bare soil, to equilibrate soil and vegetation carbon at 1850. What climate is used for this 200-y run, is it a climatology at 2000? I'm not sure that the 200-y simulations result in 'potential land surface conditions' (line 151-152), but rather an equilibrium state?

8) Line 273 states 'We therefore expect that the impact of fire on precipitation would be more significant in BGC-DV than in BGConly because fire directly influences land cover characteristics'; is this the case even though it states prior to this that Li and Lawrence (2017) found that the impact of fire on precipitation is limited? (line 251)? Perhaps they were not using dynamic vegetation, in which case it is worth making this point.

9) Line 214 states that carbon emissions from BGConly and BGC-DV are 'relatively high' but 'fall within the range of previous findings'. However BGConly emissions of 3.4 PgC are not within the range of 1.9-3.0 PgC given.

10) In a few places the text is vague and confusing, and could do with more explanation. E.g.:

Line 15: 'This study shows that inclusion of dynamic vegetation enhances carbon emissions from fire by reducing terrestrial carbon sinks; however, this effect is somewhat mitigated by the increase in terrestrial carbon sinks when dynamic vegetation is not used' – this seems like a circular argument, carbon emissions are either enhanced by DV compared to no DV, or they are reduced by no DV compared to DV

Line 193: 'Areas that experience a higher frequency of fire occurrence have larger vegetation distribution differences, which suggests that fire has an influence on vegetation mortality' – we know that fire influences vegetation mortality, isn't this is point of the paper?

Line 197 'However, there are no marked changes in the fractions of shrubs and deciduous trees in the middle of the ecological succession process with respect to the presence or absence of fire'- Specify that this is global totals, otherwise the following lines seem to contradict this

Line 198-200: 'When fire occurs in a region where shrubs grow, the ratio of shrubland is diminished, but fire increases the ratio of shrubland in regions where trees may evolve from shrubs. In the same way as shrubs, the deciduous trees are increased

or decreased due to fire' – I'm lost as to where and how fire increases shrubs and deciduous trees

Technical Corrections

1) Clarify labels for BGConly (see point 2 above)

2) Line 150 says CRU-NCEP data from (1991-2000) was used. Should this be 1901-2000 as stated in line 128?

3) Lines 206-208 there is a spare bracket

4) What is 'State vegetation', line 206?

5) The caption for figure 3 needs looking at. There are two references to BE, none to CR or BG, and the order should go from top to bottom. Also in the main text there is no mention of CR at all – I assume this is crop (see point 5 above)

6) Section 3.2 begins by saying this section considers figures 3 & 4 (line 197) but the rest of the section only refers to figures 4 and 5

7) Line 207 says 'BGC-CV' rather than BGC-DV

8) Line 221 'However, the overall NEP decrease is 2.5 Pg C y-1' – I think this should be increase if I've followed the paragraph correctly

9) Line 210 'average annual emissions are higher in BGConly (3.4 Pg)' – table 3 shows this should be 3.5 if rounding to 1d.p. as is done for BGC-DV

10) Vegetation types are not labelled in figure 5 caption

11) BE is labelled as bare ground in the caption for figure 7 but should be broadleaf evergreen. Also NEP, NPP, and Rh abbreviations should be defined fully in the caption as Net Ecosystem Production etc

12) State which correlation test is used for tables 4-7

---

## Author Response (AR1)

We thank the reviewers for their constructive comments on our manuscript. In the following paragraphs, the reviewers' comments are in black font and our point-by-point responses are in blue.

**Referee #1**

**General comments**

In this manuscript, Seo and Kim present the results of a study designed to assess the relative and interactive effects of simulating fire and dynamic vegetation on carbon and water cycling in the Community Land Model. One especially interesting finding is that fire seems to increase net ecosystem productivity, but only when dynamic vegetation is turned off. Many of the other results are not very novel, but are appropriate for Geoscientific Model Development because they add evidence supporting existing findings, and could help to interpret future CLM experiments.

This work could be valuable for the large community of researchers using CLM, as well as for global vegetation and land system modelers in general. Unfortunately, certain experimental design choices, coupled with uncertain explanation of model run setups, render parts of this manuscript impossible to confidently evaluate. I thus recommend that this paper be resubmitted with major revisions.

**Specific comments**

The spinup and transient model runs need to be much better explained. Table 1 would have been a good place to clarify things, but as it is now that table does not really give any useful information. Here's the information I would like to see in a revised Table 1, along with the gaps left by the Methods text (and a read through of the Methods for Qiu and Liu 2016):

|  | CLM4.5BGC 1850 | CLM4.5BGC 20th cent. | BGConly | BGC-DV |
|---|---|---|---|---|
| Time | ??? | 1901–2000 | 200 years | 200 years |
| Climate forcings | ??? | 1901–2000 (CRU-NCEP) | Repeated 1991–2000 (CRU-NCEP) | Repeated 1991–2000 (CRU-NCEP) |
| [CO$_2$] | ??? | ??? | ??? | ??? |
| Biogeog. shifts? | Yes? | Yes? | No (constant map) | Yes |
| Initial veg. | No (bare soil)? | Yes (as resulting from "CLM4.5BGC 1850")? | No (bare soil)? | No (bare soil)? |
| Initial soil | Uninitialized | As resulting from "CLM4.5BGC 1850" | As resulting from "CLM4.5BGC 20th cent." | As resulting from "CLM4.5BGC 20th cent." |
| Land use | ??? | ??? | At least crops? | No? |
| Fire | On? | On? | BGConly-F: On BGConly-NF: Off | BGC-DV-F: On BGC-DV-NF: Off |

>> There was a mistake in the original manuscript about the time period of climate forcing. We recycled the forcing of the 1961–2000 CRU-NCEP data, not the 1991–2001 data, for BGConly and BGC runs. This has been corrected in L159 and added to Table 1 in the revised manuscript.

>> As per reviewer's suggestion, we have added the detailed explanation of a series of different experiments in Table 1.

*Table 1: Configurations of experiments used in study.*

|  | BGC for year 1850 | BGC for 20th century | BGConly | BGC-DV |
| --- | --- | --- | --- | --- |
| Time | - | 1901–2000 | 200 yr | 200 yr |
| Climate forcing (CRU-NCEP) | Repeated 1901–1920 | 1901–2000 | Repeated for five times 1961–2000 | Repeated for five times 1961–2000 |
| [CO2] | 1850 | 1901–2000 | 2000 | 2000 |
| Biogeography shifts | No | Yes | No | Yes |
| Initial vegetation | No | From BGC year 1850 | From BGC for 20th century | No |
| Initial soil | No | From BGC year 1850 | From BGC for 20th century | From BGC for 20th century |
| Land use | 17 PFTs for 1850 | 17 PFTs for 20th century | 17 PFTs for 2000 | Simulated 15 PFTs (except crops) |
| Fire | On | On | On (BGConly-F) Off (BGConly-NF) | On (BGC-DV-F) Off (BGC-DV-NF) |

It is unclear exactly which runs were initialized with no vegetation (i.e., bare soil) because it is unclear what is being referred to by "In these simulations" on L153. The idea that the BGConly and BGC-DV runs might be initialized with suddenly bare ground is concerning; this choice could have serious carbon cycling implications by itself. This should be justified, and well.

>> While BGConly runs are initialized with vegetation, restarting from the end of the BGC for 20th century transient run, BGC-DV runs are initialized with no vegetation with soil conditions, restarting from the end of the BGC for 20th century transient run. We marked these on Table 1 to avoid confusion. Such a method for BGC-DV is commonly used to quickly stabilize the vegetation state for the year 2000 from the spun-up soil carbon state (e.g., CLM User Guide, Castillo et al. (2012) and Rauscher et al. (2015)). Furthermore, we used the final 30 years of each 200-year simulation to focus on the results after stabilization.

L148: *"In BGC-DV runs, the initial land surface state was bare ground while soil conditions were adjusted with a restart file from the transient run (i.e., BGC run for the 20th century in*

*Table 1) (Catillo et al., 2012; Raushcher et al., 2015; Qiu and Liu, 2016; Wang et al., 2016). Therefore, the vegetation state is quickly stabilized for 200 years of the BGC-DV runs since the runs restart from the spun-up soil carbon condition (i.e., after decomposition spin-up). Furthermore, the last 30 yr results of the 200 yr runs are analyzed to focus on the equilibrium states of both BGConly and BGC-DV runs."*

References
*CLM User Guide, http://www.cesm.ucar.edu/models/ccsm4.0/clm/models/lnd/clm/doc/UsersGuide/x2507.html*
*Castillo, C. K. G., Levis, S., and Thornton, P.: Evaluation of the new CNDV option of the community land model: Effects of dynamic vegetation and interactive nitrogen on CLM4 means and variability, J. Clim., 25(11), 3702–3714, doi.org/10.1175/JCLI-D-11-00372.1, 2012.*
*Rauscher, S. A., Jiang, X., Steiner, A., Williams, A. P., Michael Cai, D., and McDowell, N. G.: Sea surface temperature warming patterns and future vegetation change, J. Clim., 28(20), 7943–7961, doi.org/10.1175/JCLI-D-14-00528.1, 2015.*

If my interpretation is correct about the "Vegetation at beginning" row, how was the vegetation C present in 2000 removed for the start of the BGConly and BGCDV runs? Was it removed from the land system entirely, or was it all killed and left to decompose?

>> As in the response to the previous suggestion, BGConly runs are initialized with vegetation, restarting from the end of the BGC 20th century transient run, and BGC-DV runs are initialized with no vegetation with soil conditions, restarting from the end of the BGC 20th century transient run. We marked these on Table 1 to avoid confusion. Such a method for BGC-DV is commonly used to quickly stabilize the vegetation state for the year 2000 from the spun-up soil carbon state (e.g., CLM User Guide, Castillo et al. (2012) and Rauscher et al. (2015)). Furthermore, we used final 30 years of each 200 yr simulation to focus on the results after stabiliz of the revised manuscript.

L148: *"In BGC-DV runs, the initial land surface state was bare ground while soil conditions were adjusted with a restart file from the transient run (i.e., BGC run for the 20th century in Table 1) (Catillo et al., 2012; Raushcher et al., 2015; Qiu & Liu, 2016; Wang et al., 2016). Therefore, the vegetation state is quickly stabilized for 200 years of the BGC-DV runs because the runs restart from the spun-up soil carbon condition (i.e., after decomposition spin-up). Furthermore, the last 30 yr results of the 200 yr runs are analyzed to focus on the equilibrium states of both BGConly and BGC-DV runs."*

The use of climate forcings for 1991–2000 seems unwise. Generally, periods of at least 20–30 years are used in this sort of experiment, to better capture the full range of synoptic climate variability. It's especially egregious to use the 1990s specifically, because the 1998 ENSO event resulted in an extreme fire year.

>> There was a mistake in the original manuscript about the time period of climate forcings.

We recycled the meteorological forcing of 1961–2000 based on the CRU-NCEP data, not 1991–2001 data, for BGConly and BGC runs. This has been corrected in L159 and added to Table 1 in the revised manuscript.

It is only explained near the end of the manuscript (LL 299–300) that crops are not simulated in the BGC-DV experiments. This, along with the "CR" panels in Fig. 3 (although CR is not defined anywhere), leads me to understand that crops are simulated in the BGConly experiments. But nowhere is there any information about (a) other land uses in those experiments, (b) land uses in the spinup and transient experiments, or (c) what is used instead of cropland in the BGC-DV experiments.

>> To clarify why the crop PFTs are not included in BGC-DV runs, we have revised the following paragraph to explain the BGConly and BGC-DV runs in the revised manuscript. Furthermore, Table 1 has been added to clarify the land surface characteristics of different types of runs.

L 93: *"In addition to the SP option, CLM 4.5 can be extended using the biogeochemistry model (BGC) and dynamic vegetation model (DV); CLM simulations with BGC without DV (BGConly) and BGC with DV (BGC-DV) can be configured. BGConly simulates the carbon and nitrogen cycles in addition to biophysics and hydrology in a given distribution of vegetation PFTs (Paudel et al., 2016). In BGConly, phenological variations of LAI are simulated and whole-plant mortality is assumed as an annual mortality rate of 2% without biogeographical changes of the vegetation distribution. In contrast, BGD-DV simulates biogeographical changes in the natural vegetation distribution and mortality as well as seasonal changes of LAI (Castillo et al., 2012; 2013). A PFT can occupy a region or degenerate by competing with other PFTs, or they can coexist under various environmental factors, such as light, soil moisture, temperature, and fire (Zeng, 2010; Song and Zeng, 2013). Plant mortality in BGC-DV is determined by heat stress, fire, and growth efficiency (Rauscher et al., 2015). Note that BGC-DV does not simulate the crop PFTs because it simulates the changes in the natural vegetation only."*

>> "CR" in the caption of Figure 3 has been defined in the revised manuscript.

Figure 3: *"Percentages of land cover type (broadleaf evergreen (BE)), needleleaf evergreen (NE), deciduous (DE), shrub (SH), grass (GR), bare ground (BG) and crop (CR)) in BGC-DV-F and BGConly (the same for both BGConly-F and BGConly-NF)."*

Unfortunately, the lack of clarity with regard to the model experiment setups makes confident appraisal of the rest of the manuscript impossible. I will attempt to assess what I can, couching my comments in the necessary uncertainty.

Section 3.1 (comparing simulated burned area with GFED3) is extremely problematic. Although I'm uncertain about the specifics of the experimental design, it seems clear that the

runs are not intended as a way of reflecting reality but rather as an exploration of model mechanics. This is suggested by the use of equilibrium runs using 1991–2000 climate—a period in which the land system was certainly not in equilibrium, because of (among other factors) the continued recovery of forests in the northern hemisphere. Perhaps that's not an issue in these runs: It's possible that land use was turned off (there's no way to know, because it's not described in the Methods), but if that's the case, that's just another reason why a comparison of the model outputs to observations makes little sense. And even if one ignores all that, there's the problem that the simulations use 1991–2000 climate but the comparison is to burned area data from 1999–2011. The 1998 ENSO event resulted in an extreme fire year, which would be captured in the climate forcing (and ideally thus in the model output) but not the observational data. The third paragraph of Sect. 3.3 (LL 213–218) is problematic for the same reasons.

>> There was a mistake in the original manuscript about the time period of climate forcings. We recycled the forcing of 1961–2000 based on CRU-NCEP data, not the 1991–2001 data for BGConly and BGC runs. This has been corrected in L165 and added to Table 1 in the revised manuscript. Thus, we estimated the burned area from BGConly-F and BGC-DV-F runs by averaging the last 30 yr results for 200 yr simulations by repeating the climate forcing for 1961–2000. Therefore, the land systems of both BGConly-F and BGC-DV-F runs are equilibrated. The land use changes are not included as highlighted in Table 1. In summary, we intend to simulate the equilibrium state by repeatedly using the climate forcing.

As pointed out by the reviewer, our runs do not intend to reflect reality, but rather to explore the model process and mechanics. Thus, it is not necessary to validate the burned area against the observations, but it is still valuable to evaluate the model results using the observations.

We have therefore deleted a few unclear sentences and rewritten the paragraph to clarify our intentions and make proper comparisons with the observations. In the revised manuscript, the model results are compared with different versions of GFED datasets (GFED4 and GFED4s as well as GFED3).

L185: *"We also compare the model estimates to the satellite-based observational datasets of GFED (van der Werf et al., 2010; Giglio et al., 2013; van der Werf et al., 2017) (Figure 3). Although the model simulations are not intended to reflect the reality, but rather to understand the model mechanisms under the equilibrium states under the 1961–2000 climate forcing, it is still valuable to assess the model results using the observations. Different versions of GFED datasets provided different sized burned areas: GFED3 (van der Werf et al., 2010), GFED4 (Giglio et al., 2013), and GFED4 with small fires, i.e., GFED4s (van der Werf et al., 2017) suggest the burned area of 371 Mha yr$^{-1}$ for 1997–2009, 348 Mha yr$^{-1}$ for 1997–2011 and 513 Mha yr$^{-1}$ for 1997–2016, respectively. In comparison to the most recent data, i.e., GFED4s, both BGConly-F and BGC-DV-F runs, especially BGC-DV-F, underestimate the burned area in comparison to all three GFED datasets. Possible reasons for this underestimation in BGC-DV-F include the exclusion of agricultural fires and relatively small tree-dominated land*

*coverage." The initial model development with a BGC-DV-F type simulation (Li et al., 2012) was carried out in comparison to GFED3 (van der Werf et al., 2010) and BGC-DV-F estimated a burned area (320 Mha yr$^{-1}$) similar to that of GFED3 (i.e., 371 Mha yr$^{-1}$)."*

[Figure]

Figure 2: *"Annual burned area percentage by grid cell for CLM4.5BGC with fire (BGConly-F), CLM4.5BGCDV with fire (BGC-DV-F), and Global Fire Emission Database version 4 with small fires (GFED4s)".*

Figure 3 (land cover comparisons between BGC-DV-F and BGConly) is confounded by the fact that crops were not simulated in the BGC-DV runs. What land cover is being simulated instead? Unless there is some kind of adjustment going on, the area that should be cropland is instead in some other land cover category in BGC-DV-F.

>> As added to Table 1 of the revised manuscript, vegetation in BGC-DV runs consists of 15 different PFTs excluding crops, which can grow over the land according to the environmental conditions, including climate, weather, and soil properties. After the BGC run for the 20th century, cropland is not replaced by any other plants since BGC-DV equilibrium runs for the year 2000 are initialized with bare ground as in L212 of the revised manuscript. As in Figure 3 (percentages of land cover types), grass grows instead of crops in India and needleleaf

evergreen trees replace the agricultural land in the central part of the U.S.

L148: *"In BGC-DV runs, the initial land surface state was bare ground while soil conditions were adjusted with a restart file from the transient run (i.e., BGC run for the 20th century in Table 1) (Catillo et al., 2012; Raushcher et al., 2015; Qiu and Liu, 2016; Wang et al., 2016). Therefore, the vegetation state is quickly stabilized for 200 years of the BGC-DV runs since the runs restart from the spun-up soil carbon condition (i.e., after decomposition spin-up)."*

In Figure 5 and the discussion thereof (LL 184–192), it is unclear what is meant by "changes in the vegetation distribution." Does that refer to BGC-DV-F vs. BGConlyF, or BGC-DV-F vs. BGC-DV-NF? This makes it unclear how to interpret the results presented in the figure and text: Are we looking at an effect of including dynamic vegetation or of including fire?

>> It is the difference between BGC-DV-F and BGC-DV-NF runs to assess the impact of fires on vegetation distribution. This has been clarified in L216 as well as in Figure 5 of the revised manuscript.

L206: *"Differences in the vegetation distribution between BGC-DV-F and BGC-DV-NF in Figure 5 illustrate a nonlinear change in vegetation distribution in response to post-fire area. The changes are small in areas with minimal fire occurrence or where the burned area fraction is small (0.1–1%)."*

Figure 5: *"Differences in vegetation distribution (bare ground (BG), grass (GR), shrub (SH), deciduous (DE), broadleaf evergreen (BE), and needleleaf evergreen (NE)) ratios between BGC-DV-F and BGC-DV-NF for four burned area categories: under 0.1%, 0.1–1%, 1–10%, and greater than 10%."*

The following, in Sect. 3.4, is incomplete: "Changes in ET and runoff do not differ markedly between BGConly and BGC-DV, despite differences in the vegetation canopy and height, and soil moisture. This result could be attributed to the fact that an offline CLM was used, which does not allow for land-atmosphere interactions." It actually also indicates that including dynamic vegetation doesn't make much difference for the physiological and physical processes of the land system governing evapotranspiration and runoff.

>> As pointed out by the reviewer, we have clarified the implications of small changes in ET and runoff in the revised manuscript as follows.

L288: *"Despite the differences in soil moisture and vegetation canopy and height, changes in ET and runoff do not vary significantly between BGConly and BGC-DV. Thus, including dynamic vegetation does not impact the physiological and physical processes of evapotranspiration and runoff, respectively. However, changes in ET and runoff can be amplified in BGC-DV than in BGConly by modeling the land–atmosphere interactions with a coupled land–atmosphere model (e.g., CLM–CAM) because changes in land characteristics in*

*BGC-DV would feed back to the changes in precipitation. Therefore, the limited impact of fires on precipitation in Li and Lawrence (2017) with the coupled model would be increased by excluding dynamic vegetation in the model."*

**Other comments**

The spinup and 20th century runs were performed with CLM4.5BGC, not CLM4.5BGC-DV. What input data were used for land use and vegetation?

>> This has been clarified in Table 1 of the revised manuscript. CLM4.5BGC for the year 1850 is run using the land use data of the year 1850 and CLM4.5BGC for the 20th century is performed using the land use data for the 20th century. In terms of vegetation, BGC for the year 1850 is initialized with bare soils and BGC for the 20th century is initialized using the result of the BGC run for the year 1850.

If changes were to be made to make Sect. 3.1 justifiable (see above), why would GFED3 be used instead of the more recent GFED4, or even better, GFED4s? This could change the interpretations in Sect. 3.1, for instance, since GFED4s gives global burned area of 476 Mha/year—much closer to BGConly-F instead of BGC-DV-F.

>> As per reviewer's suggestion, we used the GFED4 and GFED4s as well as GFED3 for comparison with our results. We have revised the relevant paragraph and Figure 2 with GFED4s (the most recent version) in the revised manuscript.

L185: *"We also compare the model estimates to the satellite-based observational datasets of GFED (van der Werf et al., 2010; Giglio et al., 2013; van der Werf et al., 2017) (Figure 3). Although the model simulations are not intended to reflect the reality, but rather to understand the model mechanisms under the equilibrium states under the 1961–2000 climate forcing, it is still valuable to assess the model results using the observations. Different versions of GFED datasets provided different sized burned areas: GFED3 (van der Werf et al., 2010), GFED4 (Giglio et al., 2013), and GFED4 with small fires, i.e., GFED4s (van der Werf et al., 2017) suggest the burned area of 371 Mha $yr^{-1}$ for 1997–2009, 348 Mha $yr^{-1}$ for 1997–2011 and 513 Mha $yr^{-1}$ for 1997–2016, respectively. In comparison to the most recent data, i.e., GFED4s, both BGConly-F and BGC-DV-F runs, especially BGC-DV-F, underestimate the burned area in comparison to all three GFED datasets. Possible reasons for this underestimation in BGC-DV-F include the exclusion of agricultural fires and relatively small tree-dominated land coverage."* The initial model development with a BGC-DV-F type simulation (Li et al., 2012) was carried out in comparison to GFED3 (van der Werf et al., 2010) and BGC-DV-F estimated a burned area (320 Mha $yr^{-1}$) similar to that of GFED3 (i.e., 371 Mha $yr^{-1}$).

[Figure]

Figure 1: *"Annual burned area percentage by grid cell for CLM4.5BGC with fire (BGConly-F), CLM4.5BGCDV with fire (BGC-DV-F), and Global Fire Emission Database version 4 with small fires (GFED4s)."*

Tables 3 and 5: It is not clear what the t-tests are actually testing. Are they testing the difference of each experiment's mean difference from zero (i.e., the effects of including fire), or the difference between the two models' mean differences (i.e., the interactive effect of including dynamic vegetation and fire)?

>> We performed the t-tests for the difference between the two models' mean values (BGConly-F vs BGConly-NF and BGC-DV-F vs BGC-DV-NF). The captions of Tables 3, 5, 6 and 7 have been revised accordingly.

Table 3, 5, 6, and 7: *"their differences between the one with fire and the one without fire (i.e., BGConly-F minus BGConly-NF and BGC-DV-F minus BGC-DV-NF)"*

Throughout the paper, more effort should be made to distinguish the discussion of fire effects vs. dynamic vegetation effects.

>> We have revised the manuscript carefully to avoid any confusion with the fire effects and dynamic vegetation effects. In particular, we have clarified the confusing expressions regarding the BGC-DV and BGConly cases in Section 3.3 (Fire impact on carbon balance) and 3.4 (Fire impact on water balance). BGC-DV case means the difference between BGC-DV-F and BGC-DV-NF and BGConly case means the difference between BGConly-F and BGConly-NF. The following paragraphs are examples of these revisions.

L224: *"The impact of fires in BGConly was estimated by calculating the difference between BGConly-F and BGConly-NF, averaged over the final 30 years of each 200 yr simulation. Similarly, the impact of fires in BGC-DV was estimated by calculating the difference between BGC-DV-F and BGC-DV-NF."*

L 236: *"In addition to direct carbon emissions from fires, fire influences terrestrial carbon sinks by impacting ecosystem processes (Figure 6). Fire increases the NEP in post-fire regions in BGConly simulations (i.e., difference between BGConly-F and BGConly-NF, Figure 6a), which is consistent with the findings of the previous studies (Li et al., 2014)."*

L243: *"Simulations that ignore vegetation dynamics (i.e., the BGConly runs in this study; Li et al., 2014; Yue et al., 2015) show a global fire-induced NEP increase when comparing fire-on and fire-off runs. However, a decrease in fire-induced NEP is apparent in some regions in BGC-DV simulations (i.e., differences between BGC-DV-F and BGC-DV-NF, Figure 6b)."*

L 261: *"The impact of fires on water balance was examined by estimating the changes in runoff, evapotranspiration, and soil moisture between cases with and without fire. The differences between BGConly-F and BGConly-NF were assessed for the case without considering the vegetation dynamics and differences between BGC-DV-F and BGC-DV-F for the case considering the vegetation dynamics (Table 5 and Figure 8)."*

**Technical corrections**

L 58: Since the first FireMIP results paper has not been published, it would be more accurate to say "is evaluating" rather than "evaluated".

>> As per reviewer's suggestion, we have corrected it.

L58: *"In this respect, the Fire Model Intercomparison Project (FireMIP) evaluates the strength and weakness of each fire model by comparing the performance of different fire models and suggesting improvements for individual models (Rabin et al., 2017)."*

L 64:
– The most recent version of GFED is v4, not v3.
– Because it's the name of a specific sensor rather than a general technology, Moderate

Resolution Imaging Spectroradiometer should be capitalized.
– In addition to MODIS fire counts, GFED also considers MODIS burned area.

>> We have corrected it as follows.

L63: *"and the satellite-based Global Fire Emission Database version 3 (GFED3), which is derived from Moderate Resolution Imaging Spectroradiometer (MODIS) fire count products and burned areas, has been used to improve fire parameterizations"*

L 81: Period missing after "1.2".

>> We have corrected it.

L 301–302: "Thresholds used" should be "Thresholds are used".

>> The phrase of "Thresholds used to estimate fuel combustibility" is a subject in the sentence and thus has not been changed in the revised manuscript.

Fig. 3:
– "Plant cover" should be simply "Coverage" or something similar, because bare ground by definition has no plants.

>> We have used "land coverage" instead of "plant cover".

– "BGConly" should be "BGConly-F".

>> The same land cover is used in BGConly-F and BGConly-NF runs from the observations (i.e., MODIS) and thus we used BGConly in the caption of Figure 3

– L 512: "bare ground (BE)" should be "bare ground (BG)". CR is not defined.

>> The acronym of BG in the figure is corrected and the acronym of CR is clarified in the caption of Figure 3.

Figure 2: *"Percentages of land cover type (broadleaf evergreen (BE)), needleleaf evergreen (NE), deciduous (DE), shrub (SH), grass (GR), bare ground (BG) and crop (CR)) in BGC-DV-F and BGConly (the same for both BGConly-F and BGConly-NF)."*

Fig. 6: "Differene" should be "Difference".

>> We have corrected it in Figure 6 as follows.

[Figure]

Figure 6: *"Differences in carbon emissions (Cfe), net ecosystem production (NEP), and net ecosystem exchange (NEE) due to fires in BGConly (BGConly-F minus BGConly-NF; left column) and BGC-DV (BGC-DV-F minus BGC-DV-NF; middle column). Hashed areas indicate that the difference passed the Student's t-test at the 0.05 significance level. Latitudinal mean differences are plotted in far-right column.*

**Referee #2**

**General Comments**

This paper is evaluating the impact of incorporating fire and dynamic vegetation in the Community Land Model (CLM). The paper presents a clear description of the impact of including each of these processes by considering the change in burned area, vegetation cover, carbon balances (net ecological production and net ecosystem exchange) and water balances (evapotranspiration, runoff and soil moisture). The structure of the paper is logical and easy to follow. The method employed for this study is a set of four experiments with and without dynamic vegetation and with and without fire, which seems a valid way of testing the impact of each of these processes. However, while the paper starts well and includes a good introduction, the model description and experimental design needs more detail, and the analysis is ambiguous in several places. Specific comments on this are outlined below. The experiments here are not particularly novel, with fire and dynamic vegetation having been implemented in this model for several years as cited in the paper, but it is useful to have an evaluation of the impact of both processes as they are both important in land-surface and Earth System modelling. I recommend resubmission subject to addressing the following points.

**Specific Comments**

1) The BGConly model can be run with and without fire, and the results show that aspects of the vegetation (GPP, NPP, NEP etc, Table 3) are impacted when fire is included. But in the model description it says that the spatial distribution of PFTs is set using satellite data from MODIS and that a whole-plant mortality rate of 2% annually is assumed, rather than being determined by heat stress and fire etc. as it is in the dynamic vegetation model. So presumably the fire effects some aspects of the carbon cycle in the BGConly model, but not vegetation cover? I think this needs some clarification in the model description section – exactly what aspects are modified by including fire in the BGConly model, and what is not. It may be that this process is described in another paper, but it is necessary to understand this for the rest of this paper so it should be outlined here.

>> As per reviewer's suggestion, we have clarified what aspects are influenced by fire in BGConly as well as in BGC-DV in the revised manuscript as follows.

L96: *"In BGConly, phenological variations of LAI are simulated and whole-plant mortality is assumed as an annual mortality rate of 2% without biogeographical changes of the vegetation distribution. In contrast, BGD-DV simulates biogeographical changes in the natural vegetation distribution and mortality as well as seasonal changes in LAI (Castillo et al., 2012; 2013)."*

L118: *"Burned area also impacts the carbon and nitrogen pools of the vegetation, which are related to leaf, stem, and root; fire changes the vegetation state (e.g., LAI) and vegetation*

*height during the burning period in both BGConly and BGC-DV runs. However, the number of individual PFTs does not change in BGConly, but decreases by biomass burning in BGC-DV. In other words, individual plants are killed by fire only when the DV option is included in the model."*

2) Related to the previous point, the BGConly-F results (Figure 3) are at one point referred to as 'observations' (line 175). This may be the case in terms of vegetation cover if this is prescribed and not altered by fire, but in the rest of the paper this is one of the experimental runs being evaluated, so it needs to be clearly stated that this exactly the same as the satellite data, both in BGConly-F and BGConly-NF, and it is therefore valid to treat this as observations. The labelling throughout the text is also quite confusing which doesn't help. For example the label 'BGConly' on Figure 3b doesn't specify if this is the fire on or off run, and 'BGConly' is first described as the non-dynamic vegetation option (line 99) and then later as the impact of fire 'BGConly-F minus BGConly-NF' for figure 6 onwards (line 207).

>> We have clarified that the same land cover is used in BGConly-F and BGConly-NF runs using the observations (i.e., MODIS) in the caption of Figure 3.

[Figure]

Figure 3: *"Percentages of land cover type (broadleaf evergreen (BE), needleleaf evergreen (NE), deciduous (DE), shrub (SH), grass (GR), bare ground (BG), and crop (CR)) in BGC-DV-F and BGConly (the same for both BGConly-F and BGConly-NF)."*

>> To clarify the confusing labels, we have revised all the related text as well as the captions of Figures 6, 7, 8, and 9 in the revised manuscript as follows:

L224: *"The impact of fires in BGConly was estimated by calculating the difference between BGConly-F and BGConly-NF, averaged over the final 30 years of each 200 yr simulation. Similarly, the impact of fires in BGC-DV was estimated by calculating the difference between BGC-DV-F and BGC-DV-NF."*

L261: *"The impact of fires on water balance was examined by estimating the changes in runoff, evapotranspiration, and soil moisture between the cases with and without fire. The differences between BGConly-F and BGConly-NF were assessed for the case without considering the vegetation dynamics and differences between BGC-DV-F and BGC-DV-F for the case considering the vegetation dynamics (Table 5 and Figure 8)."*

Figure 6: *"Differences in carbon emissions ($C_{fe}$), net ecosystem production (NEP), and net ecosystem exchange (NEE) due to fire in BGConly (BGConly-F minus BGConly-NF; left column) and BGC-DV (BGC-DV-F minus BGC-DV-NF; middle column). Hashed areas indicate that difference passed the Student's t-test at the 0.05 significance level. Latitudinal mean differences are plotted in the far-right column."*

Figure 7: *"Differences in net ecosystem production (NEP), net primary productivity (NPP), and heterotrophic respiration ($R_h$)) due to fires in BGC-DV (i.e., BGC-DV-F minus BGC-DV-NF) according to the percent changes in bare ground (BG), needleleaf evergreen (NE), and grass (GR) vegetation types."*

Figure 8: *"Differences in evapotranspiration (ET) and runoff due to fires in BGConly (BGConly-F minus BGConly-NF; left column) and BGC-DV (BGC-DV-F minus BGC-DV-NF; middle column). Hashed areas indicate that the difference passed the Student's t-test at the 0.05 significance level. Latitudinal mean differences are plotted in the far-right column."*

Figure 9: *"Difference in soil moisture (%) due to fires in BGConly (i.e., BGConly-F minus BGConly-NF) and BGC-DV (i.e., BGC-DV-F minus BGC-DV-NF)."*

3) The method for calculating NEP should be included, probably before equation (3) for NEE

>> We have added the explanation of NEP in the revised manuscript.

L129: *"The terrestrial carbon balance is affected when biomass is burned. The net ecosystem exchange (NEE) can be estimated using NEP (NEP=NPP–heterotrophic respiration (Rh)) and carbon loss due to biomass burning ($C_{fe}$)."*

4) The GFED3 dataset is used here for evaluating the burned area, but there are more up to date datasets now including small fires such as GFED4.1s. Is there a reason why GFED4.1s was not used here?

>> We have used the GFEDv4 and GFEDv4s as well as GFEDv3 for comparisons with our results. We have revised the relevant paragraph and Figure 2 with GFEDv4s (the most recent version) in the revised manuscript.

L185: *"We also compare the model estimates to the satellite-based observational datasets of GFED (van der Werf et al., 2010; Giglio et al., 2013; van der Werf et al., 2017) (Figure 3). Although the model simulations are not intended to reflect the reality, but rather to understand the model mechanisms under the equilibrium states under the 1961–2000 climate forcing, it is still valuable to assess the model results using the observations. Different versions of GFED datasets provided different sized burned areas: GFED3 (van der Werf et al., 2010), GFED4 (Giglio et al., 2013), and GFED4 with small fires, i.e., GFED4s (van der Werf et al., 2017) suggest the burned area of 371 Mha yr-1 for 1997–2009, 348 Mha yr-1 for 1997–2011 and 513*

*Mha yr-1 for 1997–2016, respectively. In comparison to the most recent data, i.e., GFED4s, both BGConly-F and BGC-DV-F runs, especially BGC-DV-F, underestimate the burned area in comparison to all three GFED datasets. Possible reasons for this underestimation in BGC-DV-F include the exclusion of agricultural fires and relatively small tree-dominated land coverage." The initial model development with a BGC-DV-F type simulation (Li et al., 2012) was carried out in comparison to GFED3 (van der Werf et al., 2010) and BGC-DV-F estimated a burned area (320 Mha yr-1) similar to that of GFED3 (i.e., 371 Mha yr-1)."*

[Figure]

Figure 2: *"Annual burned area percentage by grid cells for CLM4.5BGC with fire (BGConly-F), CLM4.5BGCDV with fire (BGC-DV-F), and Global Fire Emission Database version 4 with small fires (GFEDv4s)."*

5) It seems strange that the original fire model was designed to consider vegetation dynamics (line 69) and the fire model is always run with BGC-DV (line 144), and that it simulates agricultural fire (line 105), but the DV model doesn't include crops (line 162). Presumably the agricultural fire function is only available in BGConly mode? This also needs explaining in the model description.

>> To explain that crop PFTs were not included in BGC-DV runs, we have revised the

following paragraph to explain the BGConly and BGC-DV runs in the revised manuscript. Furthermore, Table 1 has been added to clarify the land surface characteristics of different types of runs.

L 93: *"In addition to the SP option, CLM 4.5 can be extended using the biogeochemistry model (BGC) and dynamic vegetation model (DV); CLM simulations with BGC without DV (BGConly) and BGC with DV (BGC-DV) can be configured. BGConly simulates the carbon and nitrogen cycles in addition to biophysics and hydrology in a given distribution of vegetation PFTs (Paudel et al., 2016). In BGConly, phenological variations of LAI are simulated and whole-plant mortality is assumed as an annual mortality rate of 2% without biogeographical changes of the vegetation distribution. In contrast, BGD-DV simulates biogeographical changes in the natural vegetation distribution and mortality as well as seasonal changes of LAI (Castillo et al., 2012; 2013). A PFT can occupy a region or degenerate by competing with other PFTs, or they can coexist under various environmental factors, such as light, soil moisture, temperature, and fire (Zeng, 2010; Song and Zeng, 2013). Plant mortality in BGC-DV is determined by heat stress, fire, and growth efficiency (Rauscher et al., 2015). Note that BGC-DV does not simulate the crop PFTs because it simulates the changes in the natural vegetation only."*

*Table 2: Configurations of experiments used in study.*

| | BGC for year 1850 | BGC for 20th century | BGC only | BGC-DV |
|---|---|---|---|---|
| Time | - | 1901–2000 | 200 yr | 200 yr |
| Climate forcing (CRU-NCEP) | Repeated 1901–1920 | 1901–2000 | Repeated by 5 times 1961–2000 | Repeated by 5 times 1961–2000 |
| [CO2] | 1850 | 1901–2000 | 2000 | 2000 |
| Biogeography shifts | No | Yes | No | Yes |
| Initial vegetation | No | From BGC year 1850 | From BGC for 20th century | No |
| Initial soil | No | From BGC year 1850 | From BGC for 20th century | From BGC for 20th century |
| Land use | 17 PFTs for 1850 | 17 PFTs for 20th century | 17 PFTs for 2000 | Simulated 15 PFTs (except crops) |
| Fire | On | On | On (BGConly-F) Off (BGConly-NF) | On (BGC-DV-F) Off (BGC-DV-NF) |

6) ξ is the whole-plant mortality factor for each PFT (Line 120). What are the values of this factor and how is it determined?

>> We have added the details of the whole-plant mortality in the revised manuscript as follows.

L126: *"The whole-plant mortality, the rate at which plants die completely by fire, is a calibrated PFT-dependent parameter, which is 0.1 for broadleaf evergreen trees, 0.13 for needleleaf evergreen trees, 0.07 for deciduous trees, 0.15 for shrubs, and 0.2 for grasses (Li et al., 2012)."*

7) Line 134 states that 'the final surface conditions should represent those of the year 2000 after running the transient simulation'. This is fine, but line 141 states 'In these simulations, the initial global land state was bare ground. . . and soil conditions. . . were adjusted to those of the year 2000'. Does this mean the initial land state was bare ground at the start of the transient simulation, not reset at 2000? I'm not sure why you would reset vegetation at 2000 after doing a transient run, so this is probably a wording issue, but then why adjust soil conditions to 2000? I would also have expected a spin-up at the beginning considering the initial state is bare soil, to equilibrate soil and vegetation carbon at 1850. What climate is used for this 200-y run, is it a climatology at 2000? I'm not sure that the 200-y simulations result in 'potential land surface conditions' (line 151-152), but rather an equilibrium state?

>> There was a mistake in the original manuscript about the time period of the climate forcings. We have recycled the forcing of 1961–2000 based on CRU-NCEP data, not the 1991–2001 data, for BGConly and BGC runs. This has been corrected in L223 and added to Table 1 in the revised manuscript.

>> While BGConly runs are initialized with vegetation, restarting from the end of the BGC 20th century transient run, BGC-DV runs are initialized with no vegetation with soil conditions, restarting from the end of BGC 20th century transient run. We marked these on Table 1 to avoid misleading sentences. Such a method for BGC-DV is commonly used to quickly stabilize the vegetation state for the year 2000 from the spun-up soil carbon state (CLM User Guide, Castillo et al. (2012) and Rauscher et al. (2015)). Furthermore, we used final 30 years of each 200 yr simulation to focus on the results after stabilization. This point has been clarified in the revised manuscript as follows.

L148: *"In BGC-DV runs, the initial land surface state was bare ground while soil conditions were adjusted with a restart file from the transient run (i.e., BGC run for the 20th century in Table 1) (Catillo et al., 2012; Rauschcer et al., 2015; Qiu and Liu, 2016; Wang et al., 2016). Therefore, the vegetation state is quickly stabilized for 200 years of the BGC-DV runs since the runs restart from the spun-up soil carbon condition (i.e., after decomposition spin-up). Furthermore, the last 30 yr results of the 200 yr runs are analyzed to focus on the equilibrium states of both BGConly and BGC-DV runs."*

>> As suggested, we have corrected *"potential"* to *"equilibrium"* in L168.

8) Line 273 states 'We therefore expect that the impact of fire on precipitation would be more significant in BGC-DV than in BGConly because fire directly influences land cover characteristics'; is this the case even though it states prior to this that Li and Lawrence (2017) found that the impact of fire on precipitation is limited? (line 251)? Perhaps they were not using dynamic vegetation, in which case it is worth making this point.

>> As per reviewer's suggestion, we have revised the paragraph to clarify the implications of the small changes in ET and runoff and highlight the limitations of Li and Lawrence (2017).

L288: *"Despite the differences in soil moisture and vegetation canopy and height, changes in ET and runoff do not vary significantly between BGConly and BGC-DV. Thus, including dynamic vegetation does not impact the physiological and physical processes of evapotranspiration and runoff, respectively. However, changes in ET and runoff can be amplified in BGC-DV than in BGConly by modeling the land–atmosphere interactions with a coupled land–atmosphere model (e.g., CLM–CAM) because changes in land characteristics in BGC-DV would feed back to the changes in precipitation. Therefore, the limited impact of fires on precipitation in Li and Lawrence (2017) with the coupled model would be increased by excluding dynamic vegetation in the model."*

9) Line 214 states that carbon emissions from BGConly and BGC-DV are 'relatively high' but 'fall within the range of previous findings'. However BGConly emissions of 3.4 PgC are not within the range of 1.9-3.0 PgC given.

>> We have added the example of Li et al. (2012) with the estimated carbon emissions of 3.5 Pg C yr$^{-1}$.

L231: *"For example, 1997–2014 GFED4s data estimated annual direct carbon emissions as 2.3 Pg. Mouillot et al. (2006) estimated annual carbon emissions as 3.0 Pg for the end of the 20th century and the 20th century average as 2.5 Pg. Li et al. (2012) estimated the 20th century emissions as 3.5 Pg C yr-1 using the CLM3-DGVM and Li et al. (2014) and Yue et al. (2015) both estimated the 20th century emissions as 1.9 Pg C yr-1 using the CLM4.5 and ORCHIDE land surface models, respectively."*

10) In a few places the text is vague and confusing, and could do with more explanation. E.g.: Line 15: 'This study shows that inclusion of dynamic vegetation enhances carbon emissions from fire by reducing terrestrial carbon sinks; however, this effect is somewhat mitigated by the increase in terrestrial carbon sinks when dynamic vegetation is not used' – this seems like a circular argument, carbon emissions are either enhanced by DV compared to no DV, or they are reduced by no DV compared to DV.

>> We have modified the phrase to clarify the original meaning in the revised manuscript.

L15: *"Carbon emissions from fires are enhanced by reduction in NEP when vegetation dynamics are considered; however, this effect is somewhat mitigated by the increase in NEP when vegetation dynamics are not considered."*

Line 193: 'Areas that experience a higher frequency of fire occurrence have larger vegetation distribution differences, which suggests that fire has an influence on vegetation mortality' – we know that fire influences vegetation mortality, isn't this is point of the paper?

>> This statement has been removed from the original manuscript.

Line 197 'However, there are no marked changes in the fractions of shrubs and deciduous trees in the middle of the ecological succession process with respect to the presence or absence of fire'- Specify that this is global totals, otherwise the following lines seem to contradict this.

>> We have specified it as "global fractions".

L255 *"However, there are no significant changes in the global fractions of shrubs and deciduous trees in the middle of the ecological succession process with respect to the presence or absence of fires."*

Line 198-200: 'When fire occurs in a region where shrubs grow, the ratio of shrubland is diminished, but fire increases the ratio of shrubland in regions where trees may evolve from shrubs. In the same way as shrubs, the deciduous trees are increased or decreased due to fire' – I'm lost as to where and how fire increases shrubs and deciduous trees.

>> We have explained the argument by revising the sentence and pointing out the specific regions as it follows.

L 216: *"When a fire occurs in a region where shrubs grow, the ratio of shrubland is diminished (e.g., in the middle of North America in Figure 4b), but fire increases the ratio of shrubland in regions where trees grow (e.g. in the southwestern Asia in Figure 4b)."*

**Technical Corrections**

1) Clarify labels for BGConly (see point 2 above).

>> To clarify the confusing labels, we have revised the captions of Figures 6, 7, 8, and 9 in the revised manuscript.

Figure 6: *"Differences in carbon emissions ($C_{fe}$), net ecosystem production (NEP), and net ecosystem exchange (NEE) due to fires in BGConly (BGConly-F minus BGConly-NF; left column) and BGC-DV (BGC-DV-F minus BGC-DV-NF; middle column) runs. Hashed areas*

*indicate that the difference passed the Student's t-test at th 0.05 significance level. Latitudinal mean differences are plotted in the far-right column."*

Figure 7: *"Differences in net ecosystem production (NEP), net primary productivity (NPP), and heterotrophic respiration ($R_h$)) due to fires in BGC-DV (i.e., BGC-DV-F minus BGC-DV-NF) according to the percent changes in bare ground (BG), needleleaf evergreen (NE), and grass (GR) vegetation types."*

Figure 8: *"Differences in evapotranspiration (ET) and runoff due to fires in BGConly (BGConly-F minus BGConly-NF; left column) and BGC-DV (BGC-DV-F minus BGC-DV-NF; middle column) runs. Hashed areas indicate that the difference passed the Student's t-test at the 0.05 significance level. Latitudinal mean differences are plotted in the far-right column."*

Figure 9: *"Difference in soil moisture (%) due to fires in BGConly (i.e., BGConly-F minus BGConly-NF) and BGC-DV (i.e., BGC-DV-F minus BGC-DV-NF)."*

2) Line 150 says CRU-NCEP data from (1991-2000) was used. Should this be 1901- 2000 as stated in line 128?

>> As mentioned above, there was a mistake in the original manuscript about the time period of climate forcing. We recycled the forcing of the 1961–2000 CRU-NCEP data, not the 1991–2001 data, for BGConly and BGC runs. This has been corrected in L167 and added in Table 1 in the revised manuscript.

3) Lines 206-208 there is a spare bracket.

>> This part has been rewritten in the revised manuscript and thus no correction is needed.

4) What is 'State vegetation', line 206?

>> We have changed *"state vegetation"* to *"static vegetation"*.

5) The caption for figure 3 needs looking at. There are two references to BE, none to CR or BG, and the order should go from top to bottom. Also in the main text there is no mention of CR at all – I assume this is crop (see point 5 above).

>> "CR" in the caption of Figure 3 has been defined as crop in the revised manuscript.

Figure 3: *"Percentages of land cover type (broadleaf evergreen (BE)), needleleaf evergreen (NE), deciduous (DE), shrub (SH), grass (GR), bare ground (BG) and crop (CR)) in BGC-DV-F and BGConly (the same for both BGConly-F and BGConly-NF)."*

6) Section 3.2 begins by saying this section considers figures 3 & 4 (line 197) but the rest of

the section only refers to figures 4 and 5.

>> To avoid any confusion with the figure numbers, we have made the following corrections.

L198: *"The impact of fires on vegetation distribution is assessed by comparing BGC-DV-F and BGC-DV-NF simulations (Table 2 and Figures 4 and 5). Figure 4 shows the vegetation distribution of BGC-DV-NF (Figure 4a) and BGC-DV-F minus BGC-DV-NF (Figure 4b: Figure 4a minus Figure 3a);"*

7) Line 207 says 'BGC-CV' rather than BGC-DV

>> We have corrected it.

8) Line 221 'However, the overall NEP decrease is 2.5 Pg C y-1' – I think this should be increase if I've followed the paragraph correctly.

>> We have changed *"decrease"* to *"increase"*.

9) Line 210 'average annual emissions are higher in BGConly (3.4 Pg)' – table 3 shows this should be 3.5 if rounding to 1d.p. as is done for BGC-DV

>> We have corrected it.

10) Vegetation types are not labelled in figure 5 caption

>> We have clarified vegetation types in the caption of Figure 5.

Figure 5: *"Difference in vegetation distribution (bare ground (BG), grass (GR), shrub (SH), deciduous (DE), broadleaf evergreen (BE), and needleleaf evergreen (NE)) ratios between BGC-DV-F vs BGC-DV-NF for four burned area categories: under 0.1%, 0.1–1%, 1–10%, and greater than 10%."*

11) BE is labelled as bare ground in the caption for figure 7 but should be broadleaf evergreen. Also NEP, NPP, and Rh abbreviations should be defined fully in the caption as Net Ecosystem Production etc".

>> We have corrected BE as broadleaf evergreen in the caption of Figure 7.

Figure 7: *"Differences in net ecosystem production (NEP), net primary productivity (NPP), and heterotrophic respiration ($R_h$)) due to fires in BGC-DV (i.e., BGC-DV-F minus BGC-DV-NF) according to percent changes in broadleaf evergreen (BE), needleleaf evergreen (NE), and grass (GR) vegetation types."*

12) State which correlation test is used for tables 4-7.

>> As per reviewer's suggestion, we have clarified that we performed the Pearson correlation test in Table 4. In Table 5–7, we have clarified that the student's t tests were performed.

[revised manuscript text omitted]

[1] 아래로 이동함: Fire

[revised manuscript text omitted]

Spatial distributions of burned areas are compared


[revised manuscript text omitted]
…he changes in the vegetation composition within…n post-fire regions influences the above mechanisms …hows that the overall impacts…mpact of those changes in ET and runoff do…oes not differ greatly when dynamic vegetation is employed by…n the model. However,

[revised manuscript text omitted]


**Table 4:** Pearson correlation coefficients between carbon fluxes (NEP, NPP, $R_h$) and percentage changes in vegetation cover for broadleaf evergreen (BE), needleleaf evergreen (NE), deciduous (DE), shrub (SH), grass (GR), and bare ground (BG).

| | BE | NE | DE | SH | GR | BG |
|---|---|---|---|---|---|---|
| NEP | 0.84 | 0.68 | 0.34 | -0.28 | -0.80 | -0.14 |
| NPP | 0.56 | 0.44 | 0.34 | -0.30 | -0.47 | -0.35 |
| $R_h$ | -0.36 | -0.17 | -0.01 | -0.13 | 0.27 | -0.30 |

408 **Table 5:**  water budgets  ground evaporation (GE), canopy evaporation (CE), canopy transpiration (CE),
409 evapotranspiration (ET), and total runoff (RO)
410
411

|  | BGConly | | | BGC-DV | | |
|---|---|---|---|---|---|---|
|  | BGConly-F | BGConly-NF | Difference | BGC-DV-F | BGC-DV-NF | Difference |
| GE | 20.87 | 19.27 | 1.60* | 23.29 | 19.61 | 3.68* |
| CE | 15.71 | 16.39 | -0.68* | 15.62 | 16.88 | -1.26* |
| CT | 38.41 | 40.42 | -2.01* | 37.68 | 40.99 | -3.31* |
| ET | 74.99 | 76.08 | -1.09* | 76.59 | 77.48 | -0.89* |
| RO | 31.09 | 30.02 | 1.07* | 29.51 | 28.64 | 0.87* |

1412

1413

423 **Table 6 Annual mean values for LAI ($m^2$ $m^{-2}$) and vegetation height (m) and the difference between the one with fire and**
424 **the one without fire (i.e., BGConly-F minus BGConly-NF, and BGC-DV-F minus BGC-DV-NF). Asterisk (*) index indicates**
425 **that the difference passed the Student's t test at the $\alpha = 0.05$ significance level.**

| | BGConly | | | BGC-DV | | |
|---|---|---|---|---|---|---|
| | BGConly-F | BGConly-NF | Difference | BGC-DV-F | BGC-DV-NF | Difference |
| LAI | 2.13 | 2.36 | -0.23* | 2.24 | 2.62 | -0.38* |
| Height | 7.05 | 7.45 | -0.4* | 6.03 | 7.76 | -1.73* |

1426

1427


Table 7: Annual mean soil moisture (%) at each soil depth and the difference between with fire and without fire cases (i.e., BGConly-F minus BGConly-NF, and BGC-DV-F minus BGC-DV-NF). Asterisk (*) index indicates that the difference passed the Student's t test at the $\alpha = 0.05$ significance level.

| Depth | BGConly | | | BGC-DV | | |
|---|---|---|---|---|---|---|
| | BGConly-F | BGConly-NF | Difference | BGC-DV-F | BGC-DV-NF | Difference |
| 0.71 cm | 21.22 | 21.22 | 0.00* | 20.48 | 20.73 | -0.25* |
| 0.79 cm | 23.22 | 23.15 | 0.07* | 22.59 | 22.63 | -0.04* |
| 6.23 cm | 23.24 | 23.14 | 0.10* | 22.61 | 22.58 | 0.03* |
| 11.89 cm | 22.72 | 22.58 | 0.14* | 22.14 | 22.06 | 0.08* |
| 21.22 cm | 22.37 | 22.2 | 0.17* | 21.83 | 21.7 | 0.13* |
| 36.61 cm | 22.48 | 22.28 | 0.20* | 21.98 | 21.78 | 0.2* |
| 61.98 cm | 22.57 | 22.35 | 0.22* | 22.1 | 21.85 | 0.25* |
| 103.8 cm | 22.45 | 22.21 | 0.24* | 21.95 | 21.7 | 0.25* |

1439

---

## Referee Report (RR1)

**General comments**

In the original version of this manuscript, Seo and Kim presented the results of a study designed to assess the relative and interactive effects of simulating fire and dynamic vegetation on carbon and water cycling in the Community Land Model. One especially interesting finding was that fire seems to increase net ecosystem productivity, but only when dynamic vegetation is turned off. Many of the other results were not very novel, but were appropriate for Geoscientific Model Development because they add evidence supporting existing findings, and could help to interpret future CLM experiments.

The authors have done a good job of responding to reviewer comments and the revised version of the manuscript is much improved. There is still room for improvement, especially with regard to the handling of vegetation distributions in the model runs, but I recommend that this manuscript be **accepted pending minor revisions**.

**Specific comments**

There still needs to be clarification about how land use and vegetation were handled. Below is a version of Table 1 containing suggested corrections/improvements in **bold**.

| | BGC for year 1850 | BGC for 20th cent. | BGConly | BGC-DV |
|---|---|---|---|---|
| Time | — | 1901–2000 | 200 yr | 200 yr |
| Climate forcing (CRU-NCEP) | Repeated 1901–1920 | 1901–2000 | Repeated for five times 1961–2000 | Repeated for five times 1961–2000 |
| [CO$_2$] | 1850 | 1901–2000 | 2000 | 2000 |
| Biogeog. shifts? | No | Yes | No | Yes |
| Initial veg. | No | From BGC year 1850 | From BGC for 20th century | No |
| Initial soil | No | From BGC year 1850 | From BGC for 20th century | From BGC for 20th century |
| **PFTs** | **15 natural + 2 crop** | **15 natural + 2 crop** | **15 natural + 2 crop** | **15 natural** |
| Fire | On | On | On (BGConly-F) Off (BGConly-N) | On (BGC-DV-F) Off (BGC-DV-NF) |

- The "land use" row should be clarified. Based on how the authors filled it in, the row name should be "PFTs." Then the boxes should be filled with "15 natural + 2 crop" for all except the box for BGC-DV, which would have "15 natural".

- Since the "BGC for year 1850" run had no initial vegetation and no dynamic vegetation, the PFT distribution map must have come from somewhere. Where? The only explanation I see in the text is that "Initial conditions for the year 1850 equilibrium state were provided

by NCAR," but that doesn't answer the question. Presumably this run uses the Satellite Phenology option, which should be noted, since there is a paragraph spent explaining that option.

- Did the "BGC for 20th century" run use dynamic vegetation or not? There is no information about this run given in the main text, which is of course a problem. Looking at Table 1, it appears that dynamic vegetation was used (Biogeography shifts: Yes), but then later the authors state (as they also do in their reply to the other reviewer) that BGConly-F is "derived from observations". Since the initial vegetation for BGConly-F is derived from the "BGC for 20th century" run, that would seem to indicate that the latter did NOT use dynamic vegetation. I can see two ways that these two pieces of information could be reconciled:

  – If the 20th century run used an external, time-varying PFT distribution—in which case that should be noted and cited.

  – If the 20th century run used dynamic vegetation, but then the BGConly run used a set PFT distribution map from MODIS—in which case, (a) that map should be noted and cited, (b) the authors need to reconcile this with the "Initial vegetation: From BGC for 20th century" box under "BGConly" in Table 1, and (c) the authors need to explain what happened to the vegetation at the time of transition (whether it disappeared from the system entirely or was killed and left to decompose).

Other comments:

- LL148–150: This sentence should indicate whether the vegetation previously in the system was (a) killed and left to decompose or (b) removed from the system entirely in a non-C-conserving way.

- LL293–294: This sentence does not make sense in the context of this paragraph. It should be moved to the end of the previous paragraph.

**Technical corrections**

- L98: "BGD-DV" should be "BGC-DV".

- L136: "Figure 1" should be "Table 1".

- LL192–193: "in comparison to all three GFED datasets" should be deleted.

- L194: Quotation mark should be deleted.

---

## Referee Report (RR2)

General Comments

The manuscript has been much improved by the revisions and clarifications within the text in response to the referee comments. In particular, clarification of the methodology along with references, and correction of the time period of climate forcing used in the experiment I think address the main concerns from the previous version. There are still a few minor points of clarification needed as outlined below, but I recommend publication subject to these being addressed.

Specific Comments

The following points refer to the line numbering of the revised manuscript in the 'Author's Response' document which includes the tracked changes.

There is some ambiguity over the term 'land use' within the paper which needs clarification:

Where crops are included in the model, are they simulated by the model, or are they prescribed? Are they also derived from MODIS and AVHRR, or from land use data such as HYDE et al? (e.g. *Klein Goldewijk, K. , A. Beusen, M. de Vos and G. van Drecht: The HYDE 3.1 spatially explicit database of human induced land use change over the past 12,000 years, Global Ecology and Biogeography 20(1): 73-86.DOI: 10.1111/j.1466-8238.2010.00587.x., 2011* ) This is not explained in the text, and should be described in lines 233 – 351 for SP and BGC modes.

L233: Are the vegetation fractions prescribed or simulated by the model in SP mode? The text only mentions climatological data rather than vegetation cover data, but the rest of the text suggests that the SP option does use prescribed vegetation compared to simulated vegetation in the BGC-DV mode. Please clarify in the text.

Usually the term 'land use' refers to anthropogenic / agricultural land use. In Table 1 the 'land use' row would better be labelled as 'Vegetation' or 'PFTs', and should include information on whether the vegetation is simulated by the model or prescribed / derived from MODIS/AVHRR. A separate row for 'land use' including information on agricultural land use would be useful. For example:

|  | BGC for the year 1850 | BGC for the 20th century | BGConly | BGC-DV |
|---|---|---|---|---|
| Vegetation | 17 PFTs for 1850 derived from MODIS ? | Simulated / prescribed transient ? 17 PFTs for 20th century | Simulated / prescribed equilibrium ? 17 PFTs for 2000 | Simulated equilibrium 15 PFTs (without crops) |
| Agricultural land use / crops | Set at 1850, from MODIS ? | Simulated / prescribed land use change for 20th Century from ? | Set at 2000 ? | None, only natural vegetation is simulated |

L509-510 "In comparison to the burned area of BGConly-F, BGC-DV-F simulates a relatively small burned area because agricultural fires are excluded in BGC-DV-F and only natural vegetation is simulated (Castillo et al., 2012)."
Probably worth saying here as well that this is also due to fewer trees / less fuel, which is a feedback from the fire.

Technical Corrections

L911-912: "Therefore, the limited impact of fires on precipitation in Li and Lawrence (2017) with the coupled model would be increased by excluding dynamic vegetation in the model."
Should this be "including dynamic vegetation"?

L89-91: "A process-based fire parameterization of intermediate complexity has been developed and assessed within the framework of the National Center for Atmospheric Research (NCAR) the Community Earth System Model (CESM)".
This made more sense as originally written: "A process-based fire parameterization of intermediate complexity known as the Community Earth System Model (CESM) has been developed and assessed within the framework of the National Center for Atmospheric Research (NCAR)"

L216-217: "It is important to understand the individual and combined impacts of fires and vegetation distribution on water and carbon exchange; however, few studies to date have assessed this complicated global process."
Should be "these complicated global processes"

L774 "for the case without considering the vegetation dynamics and differences between BGC-DV-F and BGC-DV-F"
Should be "between BGC-DV-F and BGC-DV-NF"

---

## Author Response (AR2)

We thank the reviewers for their constructive comments on our manuscript. In the following paragraphs, the reviewers' comments are in black font and our point-by-point responses are in blue.

**Referee #1**

**General comments**

In the original version of this manuscript, Seo and Kim presented the results of a study designed to assess the relative and interactive effects of simulating fire and dynamic vegetation on carbon and water cycling in the Community Land Model. One especially interesting finding was that fire seems to increase net ecosystem productivity, but only when dynamic vegetation is turned off. Many of the other results were not very novel but were appropriate for Geoscientific Model Development because they add evidence supporting existing findings, and could help to interpret future CLM experiments.

The authors have done a good job of responding to reviewer comments and the revised version of the manuscript is much improved. There is still room for improvement, especially with regard to the handling of vegetation distributions in the model runs, but I recommend that this manuscript be accepted pending minor revisions.

**Specific comments**

There still needs to be clarification about how land use and vegetation were handled. Below is a version of Table 1 containing suggested corrections/improvements in bold.

|  | BGC for year 1850 | BGC for 20th cent. | BGConly | BGC-DV |
|---|---|---|---|---|
| Time | — | 1901–2000 | 200 yr | 200 yr |
| Climate forcing (CRU-NCEP) | Repeated 1901–1920 | 1901–2000 | Repeated for five times 1961–2000 | Repeated for five times 1961–2000 |
| [CO$_2$] | 1850 | 1901–2000 | 2000 | 2000 |
| Biogeog. shifts? | No | Yes | No | Yes |
| Initial veg. | No | From BGC year 1850 | From BGC for 20th century | No |
| Initial soil | No | From BGC year 1850 | From BGC for 20th century | From BGC for 20th century |
| **PFTs** | **15 natural + 2 crop** | **15 natural + 2 crop** | **15 natural + 2 crop** | **15 natural** |
| Fire | On | On | On (BGConly-F) Off (BGConly-N) | On (BGC-DV-F) Off (BGC-DV-NF) |

The "land use" row should be clarified. Based on how the authors filled it in, the row name should be "PFTs." Then the boxes should be filled with "15 natural + 2 crop" for all except the box for BGC-DV, which would have "15 natural".

>> As per reviewer's suggestion, we have added the detailed explanation of a series of different experiments in Table 1.

**Table 1: Configurations of the experiments used in the study**

|  | BGC for the year 1850 | BGC for the 20th century | BGConly | BGC-DV |
|---|---|---|---|---|
| Time | - | 1901–2000 | 200 yr | 200 yr |
| Climate forcing | Repeated 1901-1920 (CRU-NCEP) | 1901–2000 (CRU-NCEP) | Repeated 1961–2000 for five times (CRU-NCEP) | Repeated 1961–2000 for five times (CRU-NCEP) |
| [$CO_2$] | [1850] | [1901–2000] | [2000] | [2000] |
| Biogeography shifts | No | Yes (Prescribed with time-varying PFT distribution) | No | Yes (Simulated in DV mode) |
| Initial vegetation state | No | From BGC year 1850 | From BGC for 20th century | No |
| Initial soil | No | From BGC year 1850 | From BGC for 20th century | From BGC for 20th century |
| PFTs | 15 natural + 2 crops for 1850 based on the LUH dataset | 15 natural + 2 crops for 20th century based on the LUH dataset | 15 natural + 2 crops for 2000 based on satellite data | 15 natural (except crops) |
| Fire | On | On | On (BGConly-F) Off (BGConly-NF) | On (BGC-DV-F) Off (BGC-DV-NF) |

Since the "BGC for year 1850" run had no initial vegetation and no dynamic vegetation, the PFT distribution map must have come from somewhere. Where? The only explanation I see in the text is that "Initial conditions for the year 1850 equilibrium state were provided by NCAR," but that doesn't answer the question. Presumably this run uses the Satellite Phenology option, which should be noted, since there is a paragraph spent explaining that option.

>> We have clarified the initial conditions for the BGC for year 1850 both in the text and in "PFT" row of Table 1 in the revised manuscript (see above for Table 1).

L138: *"The BGC run for the year of 1850 was initialized with the PFT distribution from the Land Use Harmonization (LUH) transient dataset for 1850 to 2005 (Hurtt et al., 2006) to simulate the year 1850 equilibrium state, used to initialize the 20th century transient run."*

*Reference*
*Hurtt, G. C., Frolking, S., Fearon, M. G., Moore, B., Shevliakova, E., Malyshev, S., Pacala, S., and Houghton, R.: The underpinnings of land-use history:three centuries of global gridded land-use transitions, woodharvest activity, and resulting secondary lands. Glob. Change Biol. 12, 1208-1229. doi.org/10.1111/j.1365-2486.2006.01150.x, 2006.*

Did the "BGC for 20th century" run use dynamic vegetation or not? There is no information about this run given in the main text, which is of course a problem. Looking at Table 1, it appears that dynamic vegetation was used (Biogeography shifts: Yes), but then later the authors state (as they also do in their reply to the other reviewer) that BGConly-F is "derived from observations". Since the initial vegetation for BGConly-F is derived from the "BGC for 20th century" run, that would seem to indicate that the latter did NOT use dynamic vegetation. I can see two ways that these two pieces of information could be reconciled:

– If the 20th century run used an external, time-varying PFT distribution—in which case that should be noted and cited.

– If the 20th century run used dynamic vegetation, but then the BGConly run used a set PFT distribution map from MODIS—in which case, (a) that map should be noted and cited, (b) the authors need to reconcile this with the "Initial vegetation: From BGC for 20th century" box under "BGConly" in Table 1, and (c) the authors need to explain what happened to the vegetation at the time of transition (whether it disappeared from the system entirely or was killed and left to decompose).

>> "BGC for 20th century" uses an external, time-varying PFT distribution from LUH

dataset (Hurtt et al., 2006). This has been clarified both in the text and in "Biogeography shifts" and "PFT" rows of Table 1 in the revised manuscript (see above for Table 1).

L140: *"In the transient run, the amount of atmospheric carbon dioxide is increased since the onset of the Industrial Revolution in 1850 and the composition of land cover and vegetation is changed with the LUH dataset of Hurtt et al. (2006) (Vitousek et al., 1997; Pitman et al., 2004)."*

Other comments:
• LL148–150: This sentence should indicate whether the vegetation previously in the system was (a) killed and left to decompose or (b) removed from the system entirely in a non-conserving way.

>> (b) is right. We have clarified this in the revised manuscript as follows.

L150: *"In BGC-DV runs, the initial land surface state was bare ground with the vegetation previously in the system being entirely removed"*

• LL293–294: This sentence does not make sense in the context of this paragraph. It should be moved to the end of the previous paragraph.

>> We have corrected "excluding" to "including" to clarify the original meaning in the revised manuscript.

**Technical corrections**

• L98: "BGD-DV" should be "BGC-DV".

>> We have corrected it.

• L136: "Figure 1" should be "Table 1".

>> We have corrected it from "Figure 1" to "Figure 1 and Table 1".

• LL192–193: "in comparison to all three GFED datasets" should be deleted.

>> We have deleted it.

• L194: Quotation mark should be deleted

>> We have deleted it.

**Referee #2**

**General comments**

The manuscript has been much improved by the revisions and clarifications within the text in response to the referee comments. In particular, clarification of the methodology along with references, and correction of the time period of climate forcing used in the experiment I think address the main concerns from the previous version. There are still a few minor points of clarification needed as outlined below, but I recommend publication subject to these being addressed.

**Specific Comments**

The following points refer to the line numbering of the revised manuscript in the 'Author's Response' document which includes the tracked changes.

There is some ambiguity over the term 'land use' within the paper which needs clarification:

Where crops are included in the model, are they simulated by the model, or are they prescribed? Are they also derived from MODIS and AVHRR, or from land use data such as HYDE et al? (e.g. Klein Goldewijk, K. , A. Beusen, M. de Vos and G. van Drecht: The HYDE 3.1 spatially explicit database of human induced land use change over the past 12,000 years, Global Ecology and Biogeography 20(1): 73-86.DOI: 10.1111/j.1466-8238.2010.00587.x., 2011 ) This is not explained in the text, and should be described in lines 233 – 351 for SP and BGC modes.

>> The crop fractions in the gridcell are prescribed in both SP and BGC modes based on the merged dataset of the MODIS-derived land cover product and the GLC2000 data set (Ramankutty et al., 2008). This has been clarified in the revised manuscript as follows.

LL 90: *"Crop is also prescribed based on the merged dataset of the MODIS-derived land cover product and the global land cover in 2000 (GLC2000) (Ramankutty et al., 2008)."*

|  | BGC for the year 1850 | BGC for the 20th century | BGConly | BGC-DV |
|---|---|---|---|---|
| Vegetation | 17 PFTs for 1850 derived from MODIS ? | Simulated / prescribed transient ? 17 PFTs for 20th century | Simulated / prescribed equilibrium ? 17 PFTs for 2000 | Simulated equilibrium 15 PFTs (without crops) |
| Agricultural land use / crops | Set at 1850, from MODIS ? | Simulated / prescribed land use change for 20th Century from ? | Set at 2000 ? | None, only natural vegetation is simulated |

>> As per reviewer's suggestion, we have clarified the land use of the different simulations in "Biogeography shifts" and "PFT" rows of Table 1 in the revised manuscript as follows.

**Table 1: Configurations of the experiments used in the study**

| | BGC for the year 1850 | BGC for the 20th century | BGConly | BGC-DV |
|---|---|---|---|---|
| Time | - | 1901–2000 | 200 yr | 200 yr |
| Climate forcing | Repeated 1901-1920 (CRU-NCEP) | 1901–2000 (CRU-NCEP) | Repeated 1961–2000 for five times (CRU-NCEP) | Repeated 1961–2000 for five times (CRU-NCEP) |
| [CO$_2$] | [1850] | [1901–2000] | [2000] | [2000] |
| *Biogeography shifts* | *No* | *Yes (Prescribed with time-varying PFT distribution)* | *No* | *Yes (Simulated in DV mode)* |
| Initial vegetation state | No | From BGC year 1850 | From BGC for 20th century | No |
| Initial soil | No | From BGC year 1850 | From BGC for 20th century | From BGC for 20th century |
| *PFTs* | *15 natural + 2 crops for 1850 based on the LUH dataset* | *15 natural + 2 crops for 20th century based on the LUH dataset* | *15 natural + 2 crops for 2000 based on satellite data* | *15 natural (except crops)* |
| Fire | On | On | On (BGConly-F) Off (BGConly-NF) | On (BGC-DV-F) Off (BGC-DV-NF) |

L509-510 "In comparison to the burned area of BGConly-F, BGC-DV-F simulates a relatively small burned area because agricultural fires are excluded in BGC-DV-F and only natural vegetation is simulated (Castillo et al., 2012)." Probably worth saying here as well that this is also due to fewer trees / less fuel, which is a feedback from the fire.

>> This point has been added in the revised manuscript as follows.

LL 176: *"In comparison to the burned area of BGConly-F, BGC-DV-F simulates a relatively small burned area because agricultural fires are excluded in BGC-DV-F and only natural vegetation is simulated (Castillo et al., 2012) as well as because fewer trees and thus less fuels, feed backed from fire, are simulated in BGC-DV-F than in BGConly-F."*

**Technical Corrections**

L911-912: "Therefore, the limited impact of fires on precipitation in Li and Lawrence (2017) with the coupled model would be increased by excluding dynamic vegetation in the model."
Should this be "including dynamic vegetation"?

>> We have corrected it from "including" to "excluding".

L89-91: "A process-based fire parameterization of intermediate complexity has been developed and assessed within the framework of the National Center for Atmospheric Research (NCAR) the Community Earth System Model (CESM)".
This made more sense as originally written: "A process-based fire parameterization of intermediate complexity known as the Community Earth System Model (CESM) has been developed and assessed within the framework of the National Center for Atmospheric Research (NCAR)"

>> A process-based fire model is included in the NCAR CESM framework, one of earth system models, not a fire model. We therefore keep the original sentence in the revised manuscript.

L216-217: "It is important to understand the individual and combined impacts of fires and vegetation distribution on water and carbon exchange; however, few studies to date have assessed this complicated global process." Should be "these complicated global processes"

>> As per reviewer's suggestion, we have revised it.
.

[revised manuscript text omitted]